# Gap junctions deliver malonyl-CoA from soma to germline to support embryogenesis in *Caenorhabditis elegans*

**Todd A Starich[1]\*, Xiaofei Bai[2], David Greenstein[1]\***

[1]Department of Genetics, Cell Biology and Development, University of Minnesota, Minneapolis, United States; [2]National Institute of Diabetes and Digestive and Kidney Diseases, National Institutes of Health, Bethesda, United States

**Abstract** Gap junctions are ubiquitous in metazoans and play critical roles in important biological processes, including electrical conduction and development. Yet, only a few defined molecules passing through gap junction channels have been linked to specific functions. We isolated gap junction channel mutants that reduce coupling between the soma and germ cells in the *Caenorhabditis elegans* gonad. We provide evidence that malonyl-CoA, the rate-limiting substrate for fatty acid synthesis (FAS), is produced in the soma and delivered through gap junctions to the germline; there it is used in fatty acid synthesis to critically support embryonic development. Separation of malonyl-CoA production from its site of utilization facilitates somatic control of germline development. Additionally, we demonstrate that loss of malonyl-CoA production in the intestine negatively impacts germline development independently of FAS. Our results suggest that metabolic outsourcing of malonyl-CoA may be a strategy by which the soma communicates nutritional status to the germline.

**\*For correspondence:**
stari001@umn.edu (TAS);
green959@umn.edu (DG)

**Competing interests:** The authors declare that no competing interests exist.

## Introduction

Gap junctions are clusters of intercellular channels allowing direct passage of small molecules (<1000–2500 daltons) between coupled cells. They are widespread in multicellular organisms, although different gene families encode the protein subunits in invertebrates (innexins) and vertebrates (primarily connexins) (*Skerrett and Williams, 2017*). While the capability of small molecules to pass through gap junctions can be shown readily in cultured cells, identification of essential molecules that must pass through channels in vivo is challenging. Additionally, disruption of gap junction coupling can affect cell adhesion (*Meyer et al., 1992*), and resultant defects in intercellular communication may not necessarily relate to the passage of molecules through junctional channels.

In *Caenorhabditis elegans*, germ cells are coupled throughout their development to the somatic gonad by gap junctions (*Starich et al., 2014*). Germ cells progress in assembly-line fashion through the early stages of meiosis (*Figure 1A*). Gonadal gap junctions have been implicated in germ cell proliferation, oocyte growth, regulation of oocyte meiotic maturation, early embryogenesis and sperm guidance (*Govindan et al., 2006*; *Whitten and Miller, 2007*; *Nadarajan et al., 2009*; *Edmonds et al., 2011*; *Starich et al., 2014*). Octameric gap junction hemichannels (half-channels) in the somatic gonad are comprised of the recently duplicated innexins INX-8 and INX-9; apposing germ cell hemichannels are heteromeric and consist of INX-14 in conjunction with either INX-21 or INX-22 (*Figure 1—figure supplement 1*). Germ cells fail to proliferate in *inx-8(0) inx-9(0)* mutants, at a point upstream of, or parallel to, GLP-1/Notch activation of germ cells by Delta class ligands LAG-2 and APX-1. These ligands are produced in the distal tip cell (DTC) that provides the proliferative stem cell niche (*Austin and Kimble, 1987*; *Austin and Kimble, 1989*; *Henderson et al., 1994*; *Nadarajan et al., 2009*).

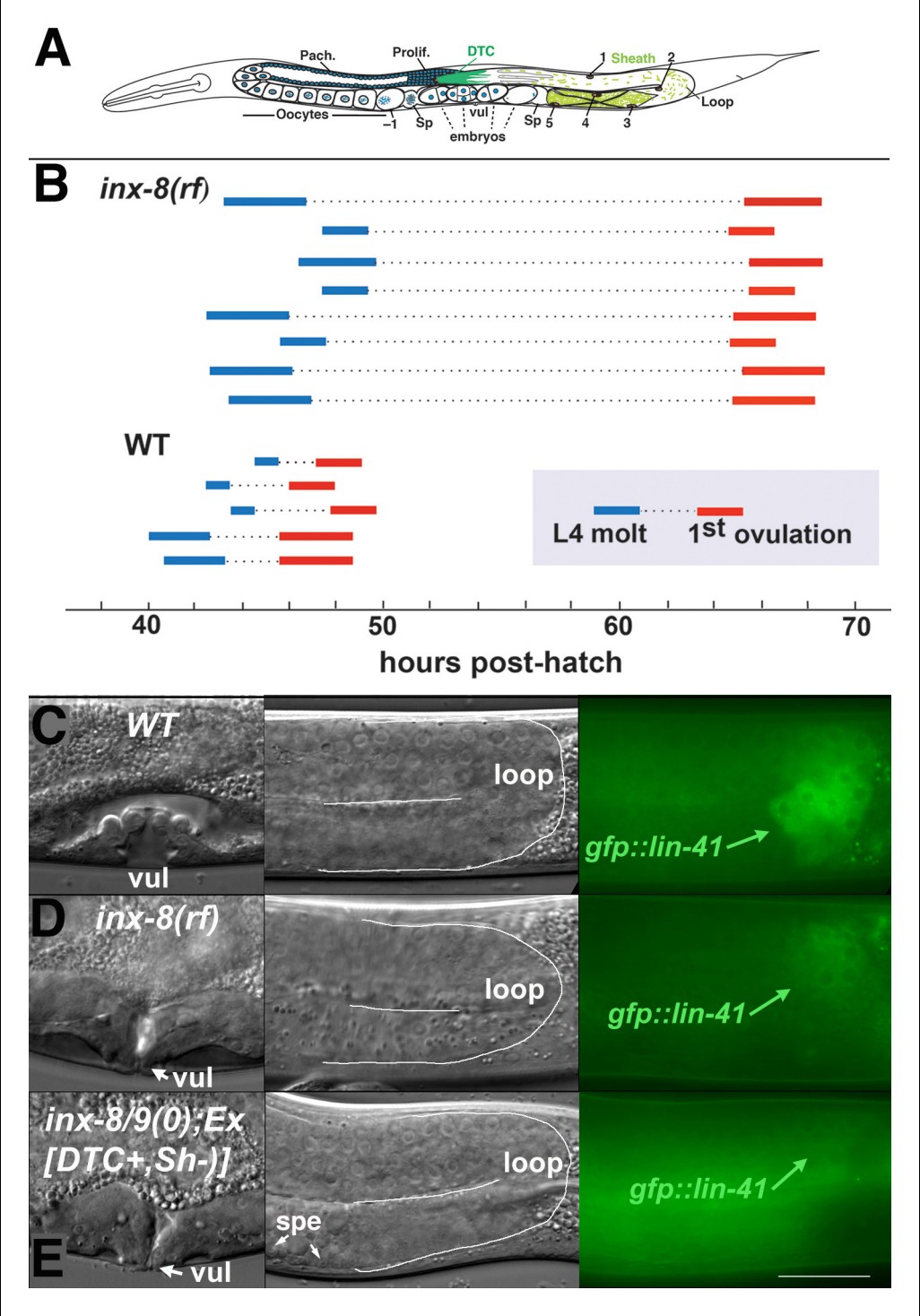

**Figure 1.** Ovulation and oocyte development are delayed in *inx-8(rf)*. (**A**) Diagram of adult *C. elegans* gonad. Posterior arm shows the positions of the somatic distal tip cell (DTC) and five pairs of sheath cells (nuclei are numbered). Anterior arm shows the underlying germline, approximate regions of germ cell proliferation (Prolif.), meiotic pachytene stage (Pach.), and developing oocytes. Embryos typically undergo some cell division in the uterus before being laid. Sp, spermatheca; vul, vulva. (**B**) Compared to the wild type (N2), ovulation in *inx-8(tn1513 tn1555rf) inx-9(ok1502null)* mutants (a.k.a. *inx-8[rf]*) is delayed ~18 hr. Bars representing 1–3 hr windows indicate approximate time of the fourth larval stage (L4 to adult) molt, followed by time of appearance of the first fertilized embryo ovulated into the uterus. (**C–E**) Onset of *gfp::lin-41* expression as a marker of oocyte development is delayed when sheath–germline gap junction coupling is perturbed. Panels show DIC and fluorescent images from

*Figure 1 continued on next page*

*Figure 1 continued*

individual worms of vulval development (left), loop region of gonad arm (middle), and expression of *gfp::lin-41* (right). (C) Wild-type hermaphrodite expressing *gfp::lin-41* by the approximate L4.7 sub-stage of vulval development (vul). (D) *inx-8(rf)* hermaphrodite does not express *gfp::lin-41* until after vulval development is complete but before the L4-to-adult molt. (E) Null *inx-8(tn1474) inx-9(ok1502)* hermaphrodite, rescued for *inx-8::gfp* expression in the DTC with *Ex[lag-2p::inx-8::gfp]*—a.k.a. *inx-8/9 (0); Ex[inx-8(DTC+, Sh–)]*—shows a similar delayed onset of *gfp::lin-41* expression as *inx-8(rf)*. spe, sperm. Bar, 20 µm.

The online version of this article includes the following source data and figure supplement(s) for figure 1:

**Source data 1.** *inx-8(rf)* exhibits reduced brood size and interacts with FA synthesis pathway genes.
**Source data 2.** Brood counts for *Figure 1*.
**Source data 3.** Reduced soma-germline gap junction coupling delays gametogenesis.
**Figure supplement 1.** Predicted topological sites of amino acids mutated in *inx-8(tn1513 tn1555) inx-9(0)*.

Germ cell proliferation can be rescued in *inx-8(0) inx-9(0)* mutants to half the level of wild type by driving *inx-8* expression in the DTC with the *lag-2* promoter (*Starich et al., 2014*). In these animals, gap junctions between soma and germline are established in the distal gonad arm, but ostensibly not in the proximal arm. Many resultant embryos arrest prior to morphogenesis and exhibit defects in cytokinesis, polar body extrusion, and eggshell development, including loss of the permeability barrier. These phenotypes resemble the Pod (Polarity and Osmotic sensitivity Defect) class of mutant embryos (*Tagawa et al., 2001*).

Disruption of genes required for fatty acid synthesis (FAS) can give rise to Pod embryos (*Rappleye et al., 1999*; *Rappleye et al., 2003*). In *C. elegans*, the *pod-2* gene encodes acetyl-CoA carboxylase (ACC) Type 1, which produces the malonyl-CoA subunits used for fatty acid chain elongation. *emb-8* encodes an NADPH-cytochrome P450 oxidoreductase that potentially modifies fatty acids. Wild-type one-cell embryos redistribute proteins to establish polarity with a resultant asymmetric first cell division. Conditional mutant alleles and RNAi silencing of both *pod-2* and *emb-8* were shown to produce high percentages of embryos that fail to properly establish polarity and divide symmetrically at the first division (*Rappleye et al., 2003*). *fasn-1* (fatty acid synthase) uses malonyl-CoA to synthesize C-16 palmitate, which can be further modified to supply a wide array of fatty acids. RNAi silencing of *fasn-1* similarly disrupted polarity establishment of one-cell embryos. In addition to loss of polarity, the osmotic barrier associated with eggshell synthesis is defective in all these embryos (*Rappleye et al., 2003*). RNAi silencing of a pair of highly similar cytochrome P450s (*pod-7* and *pod-8*) also gives rise to Pod embryos (*Benenati et al., 2009*). These P450s are implicated in hydoxylation of fatty acids. A lipid extract from *pod-7/8(+)* embryos, but not *emb-8* mutant embryos, can rescue the Pod phenotype resulting from *pod-7/8(RNAi)*, suggesting *pod-7/8* and *emb-8* may operate in the same pathway (*Benenati et al., 2009*). *pod-2*/ACC requires post-translational covalent attachment of biotin for enzymatic activity, and *bpl-1* (biotin protein ligase-1) is the *C. elegans* ortholog of the necessary holocarboxylase synthetase (*Watts et al., 2018*). Fatty acid precursors obtained from the bacterial diet enable *bpl-1* homozygous mutants from *+/bpl-1* mothers to develop to adulthood. However, resultant homozygous *bpl-1* embryos exhibit a Pod phenotype, and it was shown this is due to an early embryonic requirement for de novo fatty acid synthesis, especially for polyunsaturated fatty acids (PUFAs) (*Watts et al., 2018*).

Together these results firmly establish a requirement for fatty acid synthesis in determining polarity, implementing proper cytokinesis, and synthesizing an eggshell with an intact permeability barrier. However, mutations in non-FAS genetic pathways can also result in Pod embryos, including those affecting the APC/separase pathway (*Rappleye et al., 2002*) and genes more specifically involved in eggshell synthesis (*Zhang et al., 2005*; *Johnston et al., 2006*; *Olson et al., 2006*).

Molecules moving through soma-germ cell gap junctions necessary for proliferation and early embryogenesis are unknown. Because the disruption of gap junction coupling in the gonad can produce Pod-like embryos, we explored a potential relationship between FAS genes and gap junctions. We use genetic methods to show that malonyl-CoA (mal-CoA) is produced by POD-2/ACC in the soma. The integrity of INX-8-containing channels impacts the transfer of mal-CoA to the germline for use by FASN-1 even under conditions in which gap junction formation and cell adhesion appear normal. Rescue of the Pod phenotype in *pod-2* genetic mosaics indicates a somatic sheath focus of

action. Our results support a model in which mal-CoA transits from soma to germline through gap junction channels to facilitate embryogenesis.

## Results

### Gametogenesis is delayed by the *inx-8(tn1513 tn1555)* reduction-of-function (*rf*) mutation

We previously isolated (in an *inx-9* null background) a strong loss-of-function *inx-8* allele (*tn1513*) with sufficient residual function to contribute to channel formation and cell adhesion but not support germ cell proliferation (*Starich et al., 2014*). *tn1513* encodes a T239I change in the second extracellular loop of the innexin monomer, predicted to be involved in docking with hemichannels on apposing cells (*Figure 1—figure supplement 1*; *Oshima et al., 2016*). Using this allele in a genetic screen for restoration of germ cell proliferation, six *inx-8* intragenic suppressor mutations were isolated (to be described elsewhere). One of these, *tn1555*, encodes a D24N change in the amino terminus (*Figure 1—figure supplement 1*), near a site that may associate with a cytoplasmic dome structure surrounding the hemichannel outer pore (*Oshima et al., 2016*). Hermaphrodites of genotype *inx-8 (tn1513 tn1555rf) inx-9(ok1502null)*—for brevity to be referred to as *inx-8(rf)*—are predicted to produce homozygous hemichannels in the soma consisting of INX-8(T239I D24N) subunits. *inx-8(rf)* animals show partially restored germ cell numbers and fertility approximating one-third the level of wild type (*Figure 1—source data 1*; *Figure 1—source data 2*). *inx-8(rf)* is rescued by an *inx-8(+)* extrachromosomal array, and rescue of *inx-8(0) inx-9(0)* with an *inx-8(rf)::gfp* array mimics the phenotype of the suppressor mutant; therefore observed phenotypes can be attributed to *inx-8(rf)* and not to other possible background mutations.

There is an ~18 hr delay to the onset of fertilization in *inx-8(rf)* compared to the wild type, as determined by relating the timing of first ovulation to the L4 molt (L4-to-adult transition) (*Figure 1B*). This delay correlates with late expression of *gfp::lin41* in relation to L4 sub-stages of vulval development (*Mok et al., 2015*); LIN-41 is the earliest known marker of oogenesis (*Spike et al., 2014*). In the wild type at 20˚C, *gfp::lin41* appears by sub-stage L4.7 (*Figure 1C*); in *inx-8(rf)*, *gfp::lin41* is expressed after completion of vulval morphogenesis but prior to the L4 molt (*Figure 1D*; *Figure 1—source data 3*). Spermatogenesis is also delayed—wild-type sperm appear as early as the L4.7 stage, but *inx-8(rf)* sperm first appear after completion of vulval morphogenesis (*Figure 1—source data 3*). Once started, egg-laying in *inx-8(rf)* continues apace such that ~99% of progeny are laid within a 96 hr window (3144/3179 larvae; n = 25), similar to the wild type (5847/5929 larvae; n = 21).

### *inx-8(rf)* gap junction formation is more severely affected in proximal gonad arms

To examine expression of INX-8(T239I D24N) mutant innexins encoded by *inx-8(rf)*, specific antibodies were used to label somatic (anti-INX-8) or the two classes of germline (anti-INX-21 or anti-INX-22) hemichannels. In wild-type hermaphrodites, clusters of gap junction puncta form between somatic DTC or sheath cells and each underlying germ cell (*Figure 2A*). In distal arms of the wild type, loosely clustered sheath-germ cell gap junctions form more basally on germ cells; in *inx-8(rf)* distal arms, both INX-21 and INX-22 puncta tend to localize more laterally, where the sheath cells dip down between adjacent germ cells (*Figure 2B*). In the loop region (transition from meiotic pachytene to diplotene), gap junction puncta form large aggregates in wild-type animals; these are absent in *inx-8(rf)* (*Figure 2C*). In wild-type proximal arms (diakinesis), developing oocytes exhibit extensive formation of INX-22 gap junction puncta, while INX-21 puncta become faint; in *inx-8(rf)* proximal arms, INX-22 and INX-8 are strongly but diffusely expressed, with little co-localization evident (*Figure 2D*). These results suggest that there may be little gap junction coupling in *inx-8(rf)* proximal arms. We showed previously that sheath-oocyte gap junctions are endocytosed by the oocyte during ovulation (*Starich et al., 2014*). Embryos from *inx-8(0) inx-9(0)* rescued with *inx-8(rf):: gfp* show internalized GFP puncta, confirming that *inx-8(rf)* establishes at least some sheath-oocyte gap junctions in the proximal arm (*Figure 2—figure supplement 1*).

Reduced proximal arm expression of gap junctions is seen in *inx-8(0) inx-9(0); Ex[lag-2p::inx-8:: gfp]* animals—simplified to *inx-8/9 (0); Ex[inx-8(DTC+, Sh–)]*—which are rescued for germ cell

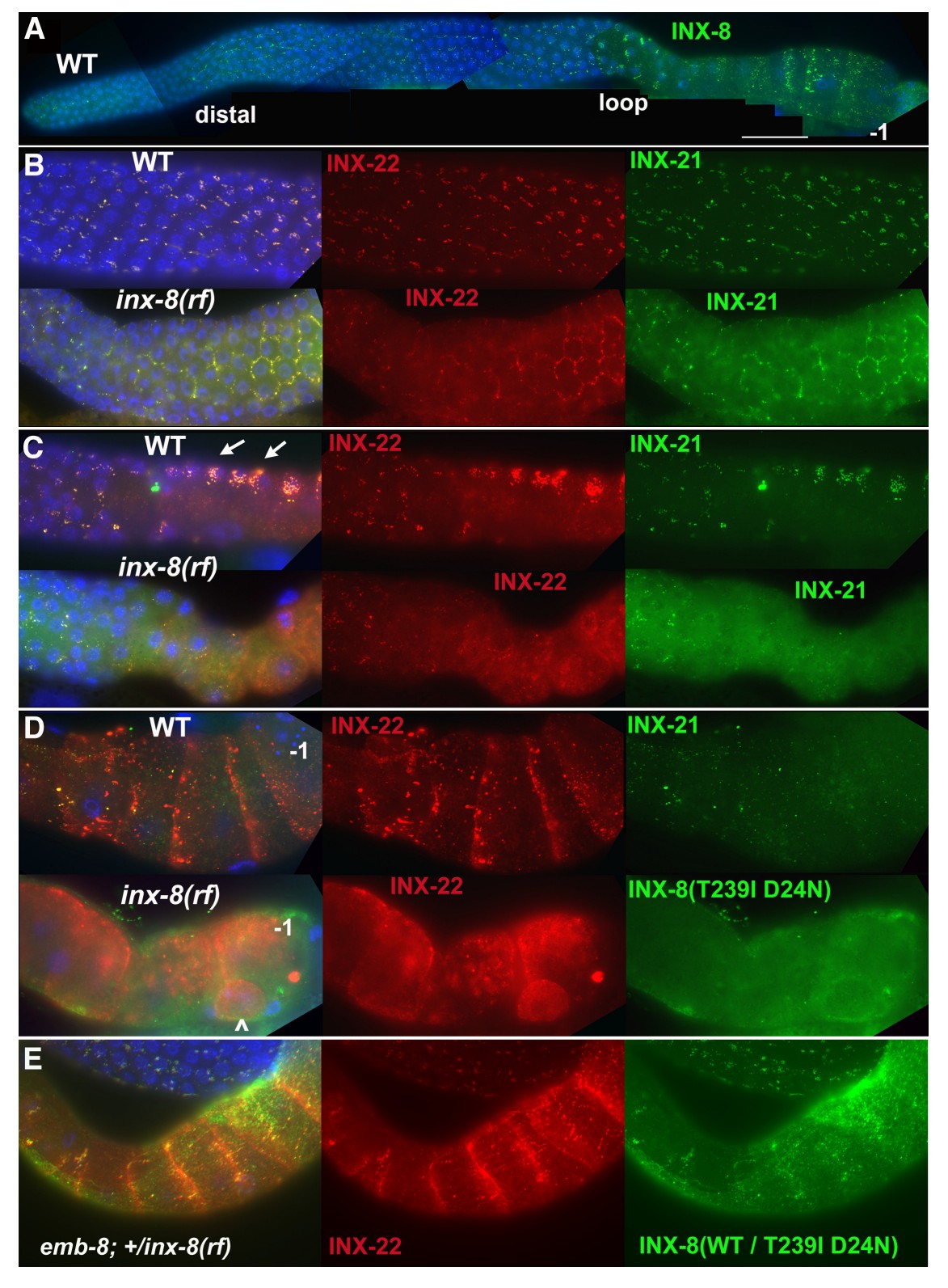

**Figure 2.** INX-8(T239I D24N) compromises gap junction formation between soma and germline. (**A**) Somatic INX-8 in wild-type (WT) gonads localizes to puncta representing soma-germline gap junctions where DTC or sheath cells overlie germ cells. (**B**) In WT distal gonad arms, gap junction puncta aggregate in small clusters associated with each germ cell; in *inx-8(rf)* distal arms, gap junction puncta localize more apically, where sheath dips down between germ cells, outlining the cell (honeycomb pattern). Localization of INX-21 and INX-22 define the two different classes of germline gap junction

*Figure 2 continued on next page*

hemichannels, both of which include INX-14 subunits. (C) At the loop region, WT gap junction puncta form large aggregates (arrows) which are absent in *inx-8(rf)*. (D) In WT proximal arms, gap junction formation with INX-22-containing hemichannels is extensive; in *inx-8(rf)*, INX-22 and INX-8(T239I D24N) are strongly expressed but show little evidence of gap junction formation. '–1' indicates the most proximal oocyte; *inx-8(rf)* hermaphrodites frequently show undersized oocytes at this position as well (<). (E) Heterozygous *+/inx-8(rf)* shows little effect on gap junction localization in an *emb-8(hc69ts)* background at 22°C. Antibodies specific to INX-8, INX-21, or INX-22 were used as indicated. DAPI-stained DNA in blue. Bar, 50 μm.

The online version of this article includes the following figure supplement(s) for figure 2:

**Figure supplement 1.** INX-8(rf)::GFP can be endocytosed by oocytes.

proliferation by driving expression of INX-8::GFP in the DTC, but not sheath (Sh), using the *lag-2* promoter (*Starich et al., 2014*). These animals produce primarily dead embryos, many with Pod characteristics (*Table 1*). In contrast to *inx-8(rf)*, these embryos do not show evidence of *inx-8::gfp* endocytosis, consistent with absence of proximal arm sheath-germline gap junction coupling (*Figure 2—figure supplement 1*). *inx-8/9 (0); Ex[inx-8(DTC+, Sh–)]* animals also show delayed onset of *gfp::lin41* expression, similar to *inx-8(rf)* (*Figure 1E*; *Figure 1—source data 3*).

We conclude that delayed gametogenesis in *inx-8(rf)* mutants arises from compromised soma-germline gap junction communication, due to altered channel properties that influence the formation and localization of gap junctions, especially in the loop and proximal arm.

## *inx-8(rf)* interacts with FAS mutants

*inx-8/9 (0); Ex[inx-8(DTC+, Sh–)]* produces Pod-like embryos, and *inx-8(rf)* shows reduced proximal arm gap junction coupling similar to *inx-8/9 (0); Ex[inx-8(DTC+, Sh–)]*. Possible genetic interactions between *inx-8(rf)* and other Pod mutants were investigated (*Figure 1—source data 1*). Conditional alleles of genes in the FAS pathway (*Figure 3A*) were used to construct double mutants with *inx-8 (rf)*. Alleles included a cold-sensitive *pod-2(ye60cs)* allele (*Rappleye et al., 2003*), and temperature-sensitive alleles of fatty acid synthase *fasn-1(g43ts)* (*Jaramillo-Lambert et al., 2015*) and *emb-8 (hc69ts)* (*Rappleye et al., 2003*). Possible interaction between *pod-2(ye60cs)* and *inx-8(rf)* was difficult to assess—*ye60cs* grown at 15°C was quite sensitive to brief handling at room temperature. At a partially restrictive temperature (22°C), *fasn-1(g43ts); inx-8(rf)* and *emb-8(hc69ts); inx-8(rf)* double mutants exhibited markedly reduced brood sizes. Similar reductions in brood size were also apparent in *emb-8(hc69ts); +/inx-8(rf)*, suggesting *emb-8(hc69ts)* might be highly dose-sensitive to *inx-8(+)* levels (*Figure 1—source data 1*).

**Table 1.** Brood sizes suggest FA synthesis influences embryo output.

| Genotype | Larvae | Dead embryos | Total (embryo output) |
|---|---|---|---|
| Wild type (N2) | 291 ± 29 (n=33) | 2 ± 2 | 293 ± 29 |
| *inx-8(rf)*[*] | 108 ± 40 (n=90) | 11 ± 6 (n=30) | 119 ± 42 |
| *inx-8/9(0); Ex[inx-8(DTC+,Sh–)]* | 2 ± 2 (n=32) | 19 ± 13 | 20 ± 14 |
| *pod-2(0); Ex[pod-2(+)]* high copy (20 ng/μl) | 164 ± 50 (n=31) | 118 ± 41 | 282 ± 59 |
| *pod-2(0); Ex[pod-2(+)]* low copy (4 ng/μl) | 91 ± 24 (n=41) | 156 ± 37 | 247 ± 50 |
| *fasn-1(0); Ex[fasn-1(+)]* | 0.05 (n=80) | 276 ± 18 (n=16) | 276 ± 18 |
| *fasn-1(0); inx-8(rf); Ex[fasn-1(+)]* | 17 ± 15 (n=47) | 43 ± 27 | 60 ± 37 |

[*]Full genotype *inx-8(tn1513 tn1555) inx-9(ok1502null)*.

The online version of this article includes the following source data for Table 1:

**Source data 1.** Brood counts for genotypes shown in *Table 1*.

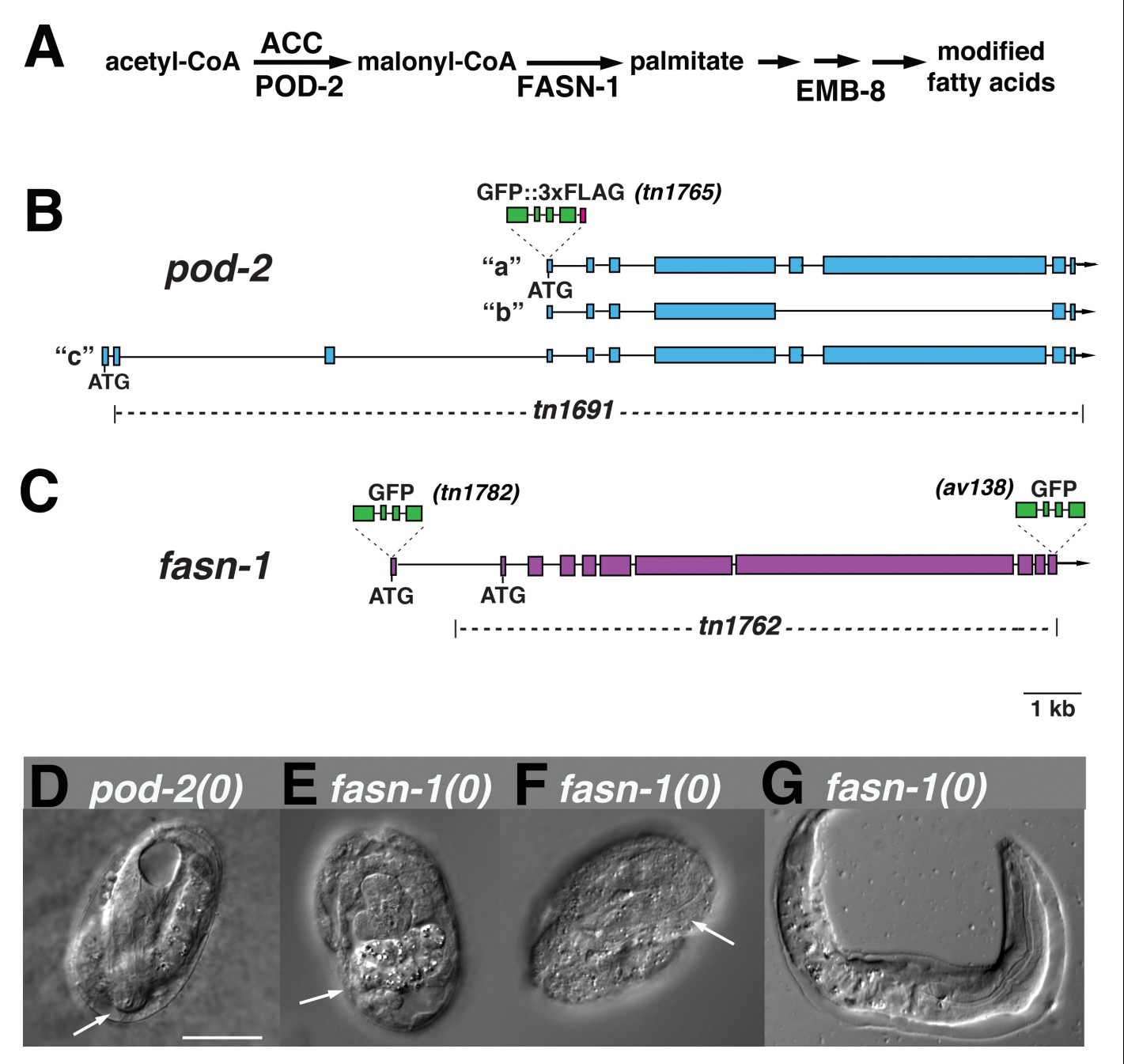

**Figure 3.** Genome editing of *pod-2* and *fasn-1*. (A) Predicted assignment of FAS gene products in a simplified FAS pathway. Targeted deletions and GFP insertions are shown for *pod-2* (B) and *fasn-1* (C). Predicted isoforms for POD-2 are indicated. (D–G) Terminal phenotypes of (D) *pod-2(0)* and (E–G) *fasn-1(0)* mutant embryos from heterozygous *+/(0)* parents. Arrows in (D) and (F) indicate differentiated anterior pharynx; arrow in (E) indicates differentiated intestinal cells that formed gut granules. Bar, 20 μm.

## Germline *fasn-1* requires somatic *pod-2*

Genetic interactions between *inx-8* and FAS genes could represent activity in the same or parallel pathways. We reasoned that if gap junctions are positioned in the FAS pathway, it might be possible to find a step within this pathway where gene function shifts from a somatic to a germline focus. *C. elegans* is particularly amenable to such an investigation because simple (highly repetitive) extrachromosomal arrays used to rescue gene function tend to express well in the soma, whereas arrays with

more sequence complexity are generally required to prevent silencing of multi-copy gene expression in the germline (*Kelly et al., 1997*).

Previous characterization of *pod-2* implied rescue by a simple array (*Rappleye et al., 2003*); we likewise found that *pod-2(ye60cs)* is rescued by simple extrachromosomal arrays at 15°C (see Materials and methods). In contrast, two fosmids encoding *emb-8*, used individually to generate simple arrays, yielded 65 transformed lines, only one of which rescued *emb-8(hc69ts)* at 25°C; the array in this line was eventually silenced. For *fasn-1*, two fosmids used to generate 93 simple array lines and 20 complex array lines failed to rescue *fasn-1(g43ts)*. These results were consistent with a model in which FAS affecting embryonic development requires *pod-2* in the soma to support a germline requirement for *fasn-1* and *emb-8*.

The special nature of a conditional allele could lead to a misinterpretation of rescue results. Because null alleles of these FAS genes had not been isolated, we generated knockouts of *pod-2 (tn1691)* and *fasn-1(tn1762)*–simplified to (*0*)–using CRISPR-Cas9 methodology (*Figure 3B and C*). *pod-2(0)* embryos from *pod-2(0)/+* parents arrest late in embryogenesis (pretzel stage; *Figure 3D*). Most *fasn-1(0)* embryos from *fasn-1(0)/+* parents exhibit signs of differentiation, and many arrest late in embryogenesis; occasional hatchlings emerge with probable hypodermal defects (*Figure 3E–G*). The terminal stages of developmental progression in both *pod-2(0)* and *fasn-1(0)* embryos derived from heterozygous parents are more advanced than classic Pod embryos, which arrest prior to signs of morphogenesis. This indicates a maternal rescue by *pod-2* and *fasn-1* of the early embryonic Pod defects.

*pod-2(0)* was rescued by simple extrachromosomal arrays, supporting probable somatic function. Using a PCR-amplified genomic fragment, we were able to rescue somatic but not germline function of *fasn-1(0)* with a simple or complex array (see Materials and methods). A genetically balanced line with a complex array, *fasn-1(0)/hT2; Ex[fasn-1(+); sur-5::gfp(+)]*, segregates healthy homozygous *fasn-1(0); Ex[fasn-1(+); sur-5::gfp(+)]* adults that lay almost exclusively dead embryos (*Table 1*). [*sur-5::gfp* is a co-injection marker expressed in somatic cell nuclei (*Yochem et al., 1998*); corresponding extrachromosomal arrays can be lost mitotically to generate mosaics—see below]]. Gonad arms appear normal, and animals lay wild-type numbers of typical Pod embryos (*Table 1*); many of these display polarity defects (4/14 symmetric first divisions) and exhibit extra nuclei in early embryos. Embryos generally arrest prior to morphogenesis (see below). We conclude this *fasn-1(+)* PCR fragment rescues somatic *fasn-1* function and is either silenced in the germline or lacks an element necessary for sufficient germline expression; in either case, a germline requirement is indicated. In sum, rescue results are consistent with a fertility requirement for *pod-2* in the somatic gonad and *fasn-1* in the germline. This conclusion is supported by genetic mosaic analysis and gene expression data described below. If *inx-8* gap junctions act in this pathway, it would position these junctions as possible mediators of mal-CoA transfer from POD-2 in the soma to FASN-1 in the germline.

## GFP::POD-2 and GFP::FASN-1 are tissue restricted

We generated endogenous, homozygous-viable, and fertile GFP-labeled alleles of *pod-2* and *fasn-1* to determine whether their patterns of localization are consistent with their predicted foci of action. The GFP moiety in *gfp::pod-2(tn1765)* is expected to label all three predicted isoforms (*Figure 3B*) (*WormBase, 2020*). GFP::POD-2 expression is strongest in the intestine, and prominent in the hypodermis, gonad sheath (especially proximal sheath), CAN neuron, and excretory duct (*Figure 4A–C*).

*fasn-1* was labeled with GFP at the amino terminus (*tn1782*) or carboxyl-terminus (*av138*) (*Figure 3C*). Both alleles exhibited similar GFP expression patterns (*Figure 5*). GFP is seen in the hypodermis, gonad sheath (esp. proximal) and excretory duct; however, intestine and CAN neuron expression are not evident (*Figure 5*). Neither GFP::POD-2 nor GFP-labeled FASN-1 is expressed in the DTC.

No germline expression of any of these GFP-tagged genes was detected, including by anti-GFP staining of dissected gonads (*Figure 4C*; *Figure 5C,F*). This was most clearly observed by examination of the distal gonad (*Figure 4C*; *Figure 5F*); however, abundant GFP::POD-2 or FASN-1::GFP expression in the proximal sheath cells made it more difficult to evaluate expression in the proximal germline. GFP::POD-2 expression in embryos first appears with development of hypodermal and intestinal cells (*Figure 4D,E*). In adults, GFP::POD-2 intestinal expression was found to decrease in response to starvation (*Figure 4F*). The earliest detectable GFP-tagged FASN-1 in embryos appears

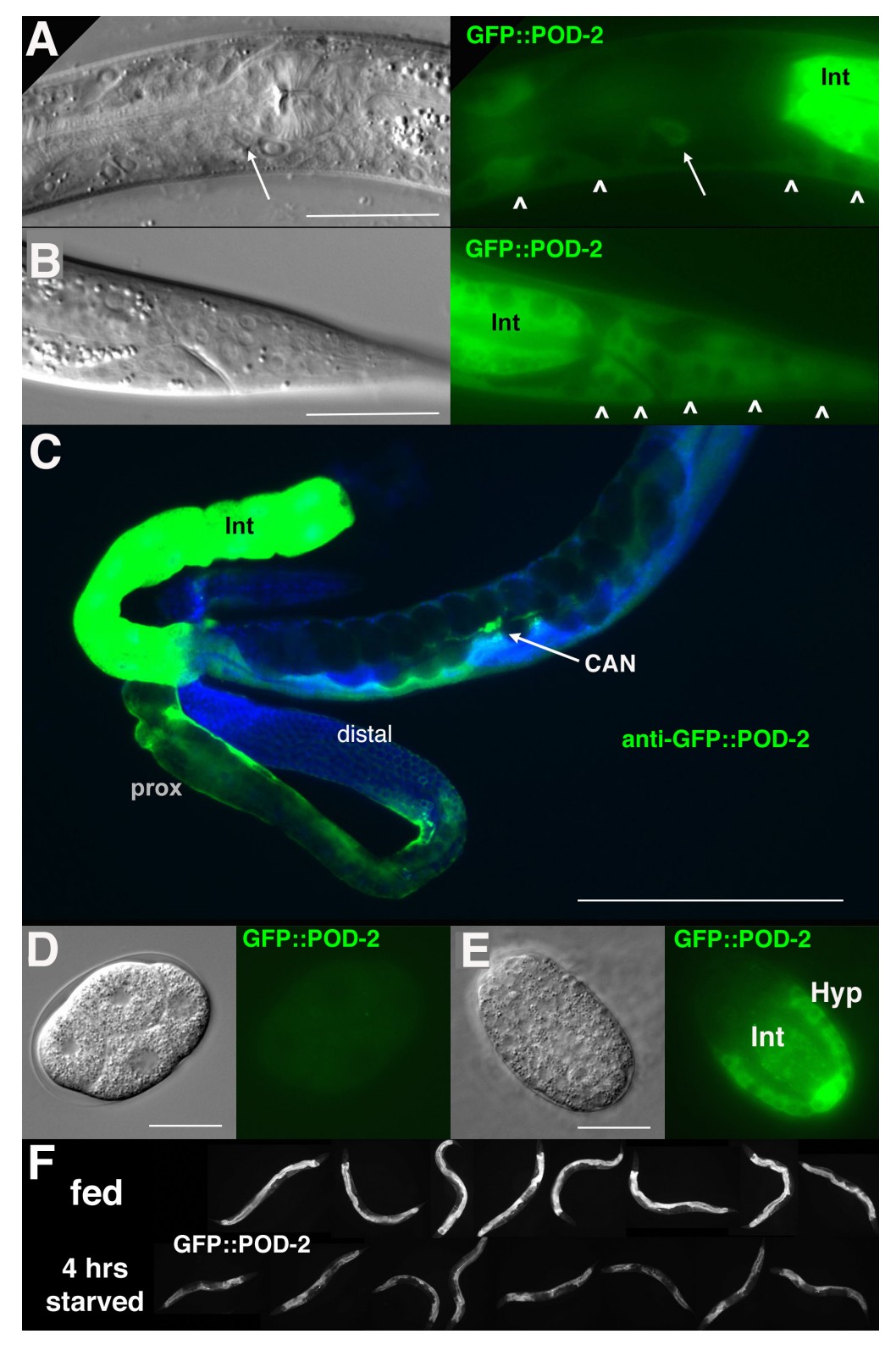

**Figure 4.** GFP::POD-2 expression. (**A**) Expression in the head includes hypodermis (carets) and excretory duct cell (arrow). Int, intestine. (**B**) Primary expression in the tail includes hypodermis (carets) and intestine (Int). (**C**) Dissected animal stained with anti-GFP antibody shows abundant intestinal expression (Int) and strong expression in proximal sheath of exposed gonad arm (prox) that diminishes in the distal arm. No germline expression is seen in the distal arm where germ cells are revealed due to the absence of sheath coverage. Arrow indicates CAN neuron, one of the few neurons

*Figure 4 continued on next page*

Figure 4 continued

essential for viability in *C. elegans*. DAPI-staining of DNA in blue. (D, E) Earliest expression in embryos is evident in developing hypodermis and intestine. (A, B, D, E) DIC images in left panels, live GFP fluorescence in right panels. Bars, 20 μm, except (C) 200 μm. (F) GFP::POD-2 is responsive to starvation. Lower row, animals briefly washed and transferred to bacteria-free NGM plates for 4 hr; upper row, animals similarly-treated but fed. Fluorescent images captured from live animals at identical exposures.

in developing hypodermis (*Figure 5D*). Expression of both genes is consistent with primarily somatic functions, while germline *fasn-1* activity may derive from low levels of maternal expression.

## INX-8 channels influence effects of overexpressed *pod-2* on downstream FAS mutations

If *inx-8* is required for transit of mal-CoA from soma to germline through gap junctions, the fertility of mutants downstream of *pod-2* in the FAS pathway could be compromised by inclusion of INX-8 (T239I D24N) into somatic hemichannels. We attempted to increase production of mal-CoA in genetically *pod-2(+)* backgrounds by additionally expressing *Ex[pod-2(+); sur-5::gfp]* simple arrays, previously shown to rescue *pod-2(0)*. Addition of *Ex[pod-2(+); sur-5::gfp]* was found to dramatically increase the brood size of *emb-8(hc69ts)* at 22°C. This increase in brood size is dependent on INX-8 hemichannel integrity—*inx-8(+)* homozygotes facilitate rescue more effectively than *+/inx-8(rf)* heterozygotes (*Figure 6A*). Importantly, although heterozygous INX-8(+)/INX-8(rf) hemichannels reduce *emb-8(hc69ts)* rescue by enhanced mal-CoA production, anti-INX-8 and anti-INX-22 co-localization in *emb-8(hc69ts); +/inx-8(rf)* dissected gonads is indistinguishable from the wild type (*Figure 2E*). Therefore, cell adhesion between soma and germline does not appear to be affected in this genetic background (*Figure 2E*).

In contrast to *emb-8(hc69ts)*, expression of *Ex[pod-2(+); sur-5::gfp]* in *fasn-1(g43ts)* significantly reduced brood size (p<0.01), and brood size reduction is more severe in homozygous *inx-8(+)* than heterozygous *+/inx-8(rf)* genetic backgrounds (ρ <0.05) (*Figure 6B*). This result likely relates to the mechanism of FASN production of fatty acids—acetyl-CoA (FA chain initiation) and mal-CoA (chain elongation) load at a common, non-selective acyl-CoA-binding site and are regarded as competitive inhibitors (*Cox and Hammes, 1983*; *Chang and Hammes, 1990*). Increasing the ratio of mal-CoA to acetyl-CoA may inhibit FASN-1 enzyme by interfering with chain initiation. Both wild-type and enhanced levels of mal-CoA delivered through wild-type INX-8 hemichannels appear to inhibit *fasn-1(g43ts)*; this mutant form of FASN may be especially sensitive to inhibition by higher levels of mal-CoA. Importantly, heterozygous *+/inx-8(rf)* channels ameliorate this inhibition (*Figure 6B*), likely by reducing mal-CoA transfer to the germline.

*inx-8(+)* hemichannels can have both a positive (*emb-8*) and a negative (*fasn-1*) genetic effect on FAS mutations which function downstream of overexpressed *pod-2(+)*. This finding is a strong indication that *inx-8* acts in the same pathway as *pod-2*, rather than a parallel pathway. These results are consistent with gap junctions acting as conveyors of mal-CoA from soma to germline, and they fit a model in which INX-8 gap junctions are positioned between *pod-2* and *fasn-1* (*Figure 6C*), but not downstream of *fasn-1* (*Figure 6D*).

## Genetic mosaic analyses associate *pod-2* sheath cell loss with the Pod phenotype

To verify a somatic *pod-2* focus of action, genetic mosaic animals derived from *pod-2(0); Ex[pod-2(+); sur-5::gfp]* were sought. 17/833 adult hermaphrodites were identified that produced dead eggs but no viable progeny; these adults were examined for cell lineage loss of *Ex[pod-2(+); sur-5::gfp]*. Animals fell into two classes: (1) eight animals lost *Ex[pod-2(+); sur-5::gfp]* from most or all gonad sheath cells and (2) nine animals lost *Ex[pod-2(+); sur-5::gfp]* from the intestine (*Figure 7A*).

Within the first class of mosaics, five animals lost the array in all sheath cells in both gonad arms. Three other animals exhibited a difference in phenotypic severity between gonad arms—embryos derived from one arm developed to later multicellular stages in utero (*Figure 7B–D*). Differences were coincident with loss of the array from all sheath cells in the more severely affected arm, versus retention of the array in a single sheath cell of the better arm. In subsequent screening, five more mosaics were identified that displayed phenotypic disparity between the two gonad arms; in one example, one arm produced defective embryos while the better arm gave rise to wild-type embryos

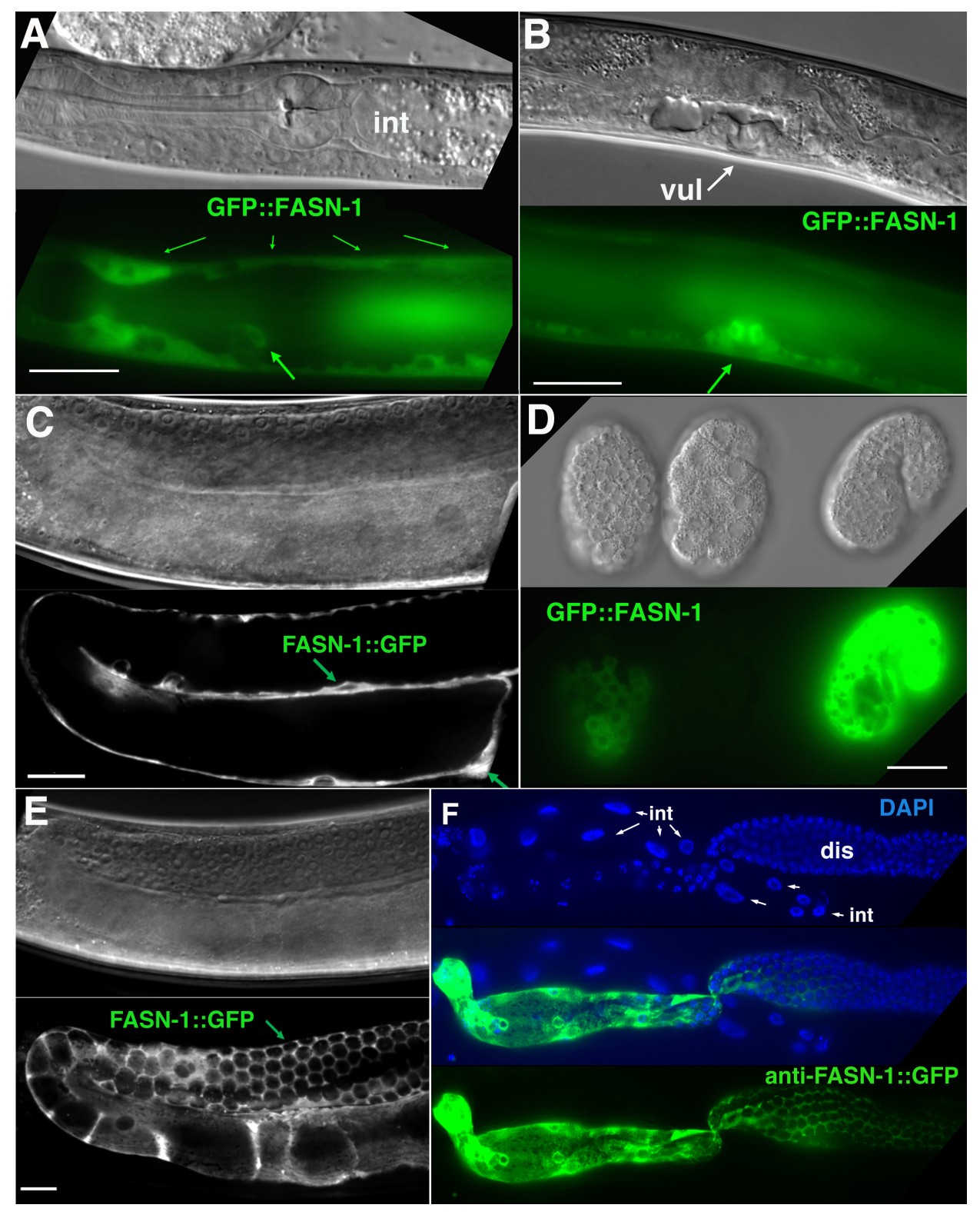

**Figure 5.** Expression of N-terminal GFP::FASN-1 or C-terminal FASN-1::GFP CRISPR-Cas9 constructs. (A) Expression is prominent throughout the hypodermis (small arrows) and excretory duct cell (larger arrow), but not intestine (int). (B) Expression in developing vulva (arrow). (C) Gonadal sheath expression, with absence of apparent germline expression. (D) Earliest expression is seen in the hypodermis of developing embryos, and hypodermal expression continues throughout development. (E) Expression is apparent in all somatic sheath cells (Sh1–5). (F) Anti-GFP antibody staining of dissected

*Figure 5 continued on next page*

*Figure 5 continued*
gonad and intestine expressing FASN-1::GFP. DAPI staining shows large nuclei (int) of intestine crossing underneath the gonad. Strong expression is detected in the proximal gonad, less so in distal gonad (dis), but no intestinal expression is apparent. (A–E) DIC images in upper panels, live GFP fluorescence in lower panels. Bars, 20 μm.

(producing $\geq$40 viable progeny). These results indicate that the focus of *pod-2(0)* rescue is autonomous to each gonad arm, consistent with the interpretation that *pod-2* expression in the somatic sheath is essential to embryonic viability.

In subsequent screening for germline ($P_4$–) mosaics, four animals were identified that maintained the array in the soma but lost the array in all embryos (*Figure 7A*). The terminal stages of embryonic development resembled those of *pod-2(0)* embryos derived from *+/pod-2(0)* parents, that is embryos developed to threefold or pretzel stages but failed to hatch (*Figure 7E–G*). These stages are more advanced than terminal stages of many dead embryos derived from parents with somatic (but not germline) loss of the rescuing array; such embryos fail to differentiate (*Figure 7H*). This early developmental arrest resembles terminal-stage Pod embryos from *fasn-1(0); Ex[fasn-1(+)]* (compare panels *H* and *I* in *Figure 7*). We conclude that somatic *pod-2(+)* supplies sufficient levels of mal-CoA to the germline not only to establish a wild-type one-cell embryo, but also to support continued development, presumably until zygotic *pod-2* expression becomes adequate for completion of embryogenesis. These studies establish that metabolites essential for embryonic development can be produced in the somatic gonad and provided to the oocyte through gap junctions.

## Intestinal loss of *pod-2*, but not *fasn-1*, results in a germline starvation-like phenotype

The other class of *pod-2(0)* genetic mosaics producing dead embryos in our original screen lost *Ex [pod-2(+); sur-5::gfp]* from the intestine (E–) (*Figure 7A*). E(–) mosaic animals were smaller than wild type, and gonad arms were severely reduced in size (*Figure 8A*). Only the most proximal oocytes showed appreciable growth. E(–) mosaic gonad arms resemble those seen in gonads undergoing the oocyte starvation response, or adult reproductive diapause (ARD), a process induced by starvation during the L4 larval stage proposed to maintain germline stem cells for repopulation of the gonad upon the resumption of feeding (*Angelo and Van Gilst, 2009*; *Seidel and Kimble, 2011*). During ARD, germline stem cells cease dividing, the gonad shrinks due to extensive apoptosis of differentiated germ cells, and growth is largely restricted to the most proximal oocyte.

The *C. elegans* intestine is enriched with lipid droplets, produces yolk lipoproteins (*Kimble and Sharrock, 1983*) and is regarded as a major site of FAS (*Watts and Ristow, 2017*). A genetic pathway in *C. elegans* required for intestinal apical polarity establishment leading to proper lumen formation has been described; genes in this pathway, including *pod-2*, encode enzymes required for glycosphingolipid synthesis (*Zhang et al., 2011*). Lipid biosynthesis is required throughout larval growth, coincident with membrane expansion. Our isolation of adult mosaics lacking *pod-2(+)* in the intestine was therefore surprising. We subsequently identified two presumptive $P_1$(–) *pod-2* mosaics, predicted to have lost *pod-2(+)* function in the intestine, somatic gonad, germline, and many muscle and hypodermal cells, among others (*Figure 7A*). The most striking phenotype distinguishing these animals from E(–) mosaics was an extremely constipated intestine (*Figure 8B*). Therefore the absence of this constipation in E(–) mosaic animals implies that loss of *pod-2(+)* from the intestine does not prevent the processing of bacteria through the intestinal lumen. (Both $P_1$ mosaics showed tissue extruded from the rectum, which may be the cause of constipation.)

Although *gfp::fasn-1(tn1782)* is not detected in the intestine, we asked if there is a functional role for *fasn-1* similar to *pod-2*. As described above, homozygous *fasn-1(0); Ex[fasn-1(+); sur-5::gfp]* progeny from heterozygous parents produce Pod embryos but exhibit no other phenotype; within this class of progeny we screened for E(–) mosaics. Ten (of 800) such animals were identified (*Figure 7A*); in stark contrast to *pod-2*, their gonad arms appeared normal. These animals displayed no overt defects outside of Pod embryo production (*Figure 8C*). Because maternal FASN-1 rescues *fasn-1(0)* homozygotes to later embryonic stages, an early intestinal function for *fasn-1* equivalent to *pod-2* could be rescued maternally. However, the surprising conclusion from this mosaic analysis is that there is no significant intestinal requirement for zygotic *fasn-1* in larval and adult stages. The

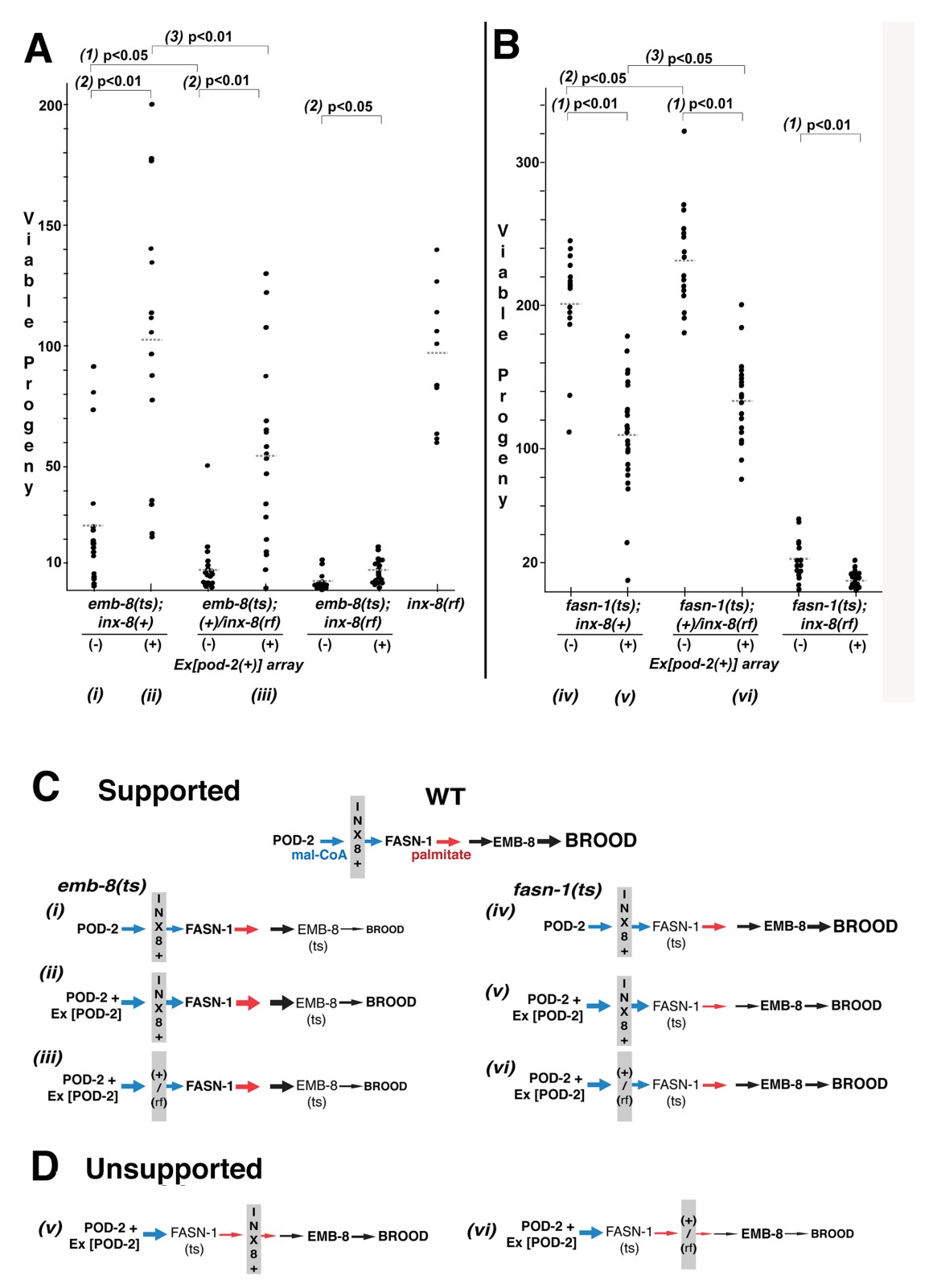

**Figure 6.** Wild-type INX-8 facilitates effects of *pod-2(+)* overexpression on conditional mutations downstream in the fatty acid synthesis pathway. Brood size was measured as the number of viable progeny produced. The balancer *mIs11* provided *inx-8(+)* for all strains listed. (A) Embryonic lethality in *emb-8(ts)* mutants at 22°C is more severe in heterozygous *+/inx-8(rf)* than *inx-8(+)* backgrounds (*1; p=0.013*). Additional *pod-2(+)* expression furnished by *Ex[pod-2(+)]* in *emb-8(ts)* significantly rescues embryonic lethality (*2; p=1.9 E-05, inx-8[+]; p=1.9 E-05, +/inx-8[rf]; p=0.022, inx-8[rf]*). Rescue of *emb-8*
*Figure 6 continued on next page*

Figure 6 continued

(ts) by Ex[pod-2(+)] is also dependent on inx-8(+) copy number (3; p=5.5 E-03). **(B)** Expression of Ex[pod-2(+)] in fasn-1(ts) mutants at 20°C reduces brood size (1; p=2.2 E-08, inx-8[+]; 1.5 E-10, +/inx-8[rf]; p=7.0 E-04, inx-8[rf]). fasn-1(ts); inx-8(+) homozygotes show greater embryonic lethality than fasn-1(ts); (+)/inx-8(rf) heterozygotes in the absence (2; p=0.026) or presence (3; p=0.026) of Ex[pod-2(+)]. Averages indicated with dotted lines. Probability was calculated using Student's two-tailed t-test. **(C)** Interpretation of genetic interaction consistent with the results from **(A)** and **(B)** in a pathway representation. Relative size of arrows corresponds to expected relative amount of product. Sheath–germline gap junctions composed of INX-8(+) or a mixture of INX-8(+) and INX-8(rf) are positioned as facilitating transfer of malonyl-CoA from soma to germline. Roman numerals correspond to genotypes indicated in **(A)** and **(B)**. **(D)** Placement of INX-8 gap junctions downstream of fasn-1 predicts brood size from genotype (vi) would be less than from (v), which is not consistent with the results observed in **(B)**. *Figure 6—source data 1* Viable progeny counts for panels **(A, B)**.

The online version of this article includes the following source data for figure 6:

**Source data 1.** Brood counts for *Figure 6*.

implication is that the primary function of mal-CoA produced in the intestine is not as substrate for intestinal FASN-1.

## Is malonyl-CoA required in other germline processes?

inx-8(rf) and inx-8/9(0); Ex[inx-8(DTC+, Sh–)] mutants share delayed oogenesis and a reduction in gap junction coupling, germ cell proliferation, and brood size (*Table 1*). inx-21(0) mutants are sterile, and inx-22(0) mutants are fertile (*Starich et al., 2014*); therefore, mal-CoA required for embryonic development must minimally transit through gap junction channels composed of INX-14/INX-21 germline hemichannels. These channels are strongly expressed throughout the distal arm, but expression wanes in the proximal arm (*Figure 2B*), a pattern somewhat incongruous with mal-CoA being required only for embryo integrity. Because inx-8/9(0); Ex[inx-8(DTC+, Sh–)] mutants appear to have no sheath–germline gap junction coupling, their associated phenotypes should represent the most severe defects possible if all gap-junctional mal-CoA transfer from sheath to germline is eliminated, that is, reduction of germ cell numbers by half, delayed gametogenesis, or very low embryo output (*Table 1*). We asked if mal-CoA levels might indeed influence phenotypes preceding embryogenesis.

Attempts to supply mal-CoA transiently by microinjection (0.5–10 mM) into inx-8/9(0); Ex[inx-8(DTC+, Sh–)] L4 gonad arms failed to ameliorate any phenotypes. Ectopic expression of GFP::POD-2 in the germline in inx-8/9(0); Ex[inx-8(DTC+, Sh–)] or inx-8(rf) genetic backgrounds also failed to improve phenotypes, including delayed gametogenesis (*Figure 7—figure supplement 1*). However, whether endogenous mal-CoA levels are altered under these conditions is uncertain.

Effects of global mal-CoA levels were indirectly examined by comparing higher or lower pod-2(+) copy number rescue of pod-2(0); numbers of dead eggs and larvae were summed to reflect total embryo output (*Table 1*). Lower copy number rescue coincided with reduced embryo output (84% of wild-type) and lower proportion of viable larvae within a brood (~37% of total embryos) (*Table 1*). Embryo output may therefore be impacted by pod-2 expression; however, somatic cells outside the gonad could be responsible for mediating this effect.

We therefore asked specifically if FAS is required within the germline for embryo production. Somatically rescued fasn-1(0); Ex[fasn-1(+); sur-5::gfp] embryo output is comparable to the wild type (94%), consistent with a minimal role for germline fasn-1 in promoting embryo output. Further, we observed that embryo output in the fasn-1(0); inx-8(rf); Ex[fasn-1(+); sur-5::gfp] double mutant is reduced to ~50% of inx-8(rf) alone (*Table 1*). One interpretation of this finding is that mal-CoA, transferred to the germline via gap junctions, functions independently of FAS to promote embryo output. This interpretation is complicated by the observation that fasn-1(0); Ex[fasn-1(+); sur-5::gfp] adult hermaphrodites produce a very small number of embryo escapers that grow to adulthood, suggesting the possibility that Ex[fasn-1(+); sur-5::gfp] retains some residual germline activity that supports embryo output.

Surprisingly, inx-8(rf) partially suppresses the embryonic lethality of fasn-1(0); Ex[fasn-1(+); sur-5::gfp] such that fasn-1(0); inx-8(rf); Ex[fasn-1(+); sur-5::gfp] strains can be maintained as viable and fertile lines with low brood sizes (*Table 1*). Possibly the smaller gonad arms of inx-8(rf) may concentrate levels of mal-CoA or FASN-1, which would be consistent with a low level of germline expression from Ex[fasn-1(+); sur-5::gfp]. Alternatively, Ex[fasn-1(+)] germline expression may be partially desilenced in this genetic background. Gene silencing by small RNAs is intimately associated with the

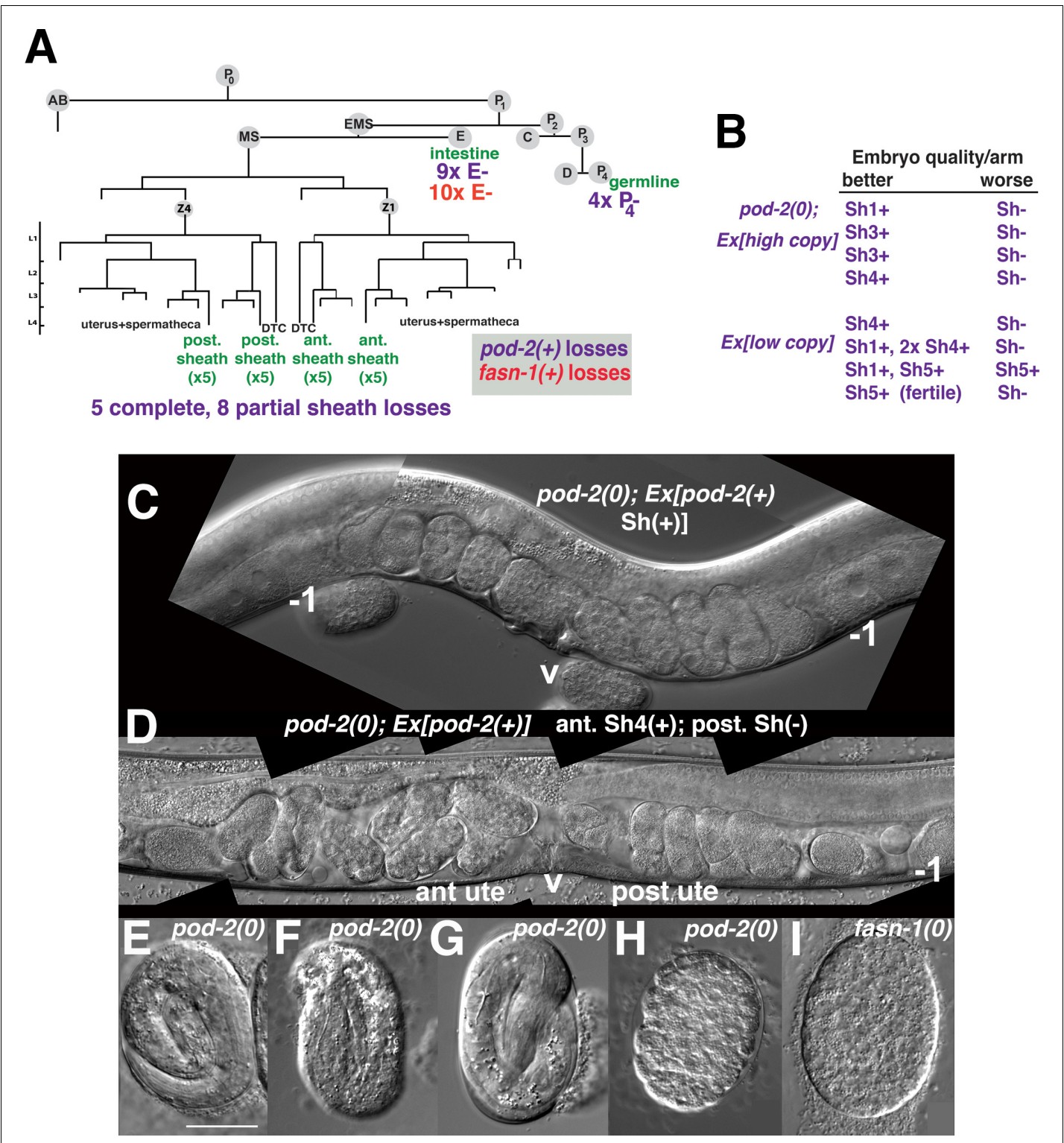

**Figure 7.** Genetic mosaic analyses of *pod-2* reveal a mal-CoA requirement in somatic gonad sheath. (A) Abbreviated cell lineage of *C. elegans*. Mosaics characterized correspond to loss of *sur-5::gfp* expression in the germline (P4), intestine (E), or somatic gonad. Five *pod-2(0); Ex[pod-2(+); sur-5:: gfp]* animals lost the array from all somatic gonad sheath cells and produced no viable embryos. (B) Eight animals showed differences in array retention in each gonad arm with phenotypic consequences. Arrays were generated by microinjecting cosmid W09B6 *[pod-2(+)]* at 4 ng or 20 ng/μl (low/high copy). (C) Uterus of non-mosaic *pod-2(0); Ex[pod-2(+)*—high copy] hermaphrodite; developmental progression of embryos from each gonad arm is comparable. (D) Mosaic *pod-2(0); Ex[pod-2(+)*—high copy] with array lost from posterior gonad arm sheath but retained in a single Sh4 cell of anterior

*Figure 7 continued on next page*

*Figure 7 continued*

arm. Anterior arm embryos have more distinct eggshells and develop to later stages. (E–H) Embryonic development depends on somatic levels of mal-CoA. Shown are terminal-stage *pod-2(0)* embryos derived from parents of designated genotypes. (E) Pretzel-stage embryo from *+/pod-2(0)*. (F) Threefold stage embryo from *pod-2(0); Ex[pod-2(+)—low copy]*, with array maintained in soma but lost in the germline (P$_4$–). (G) Pretzel-stage embryo from P$_4$(–) mosaic *pod-2(0); Ex[pod-2(+)—high copy]*. (H) Embryo from mosaic *pod-2(0); Ex[pod-2(+)—low copy]* with uncharacterized somatic loss of array, but array retained in the germline (P$_4$+). This parent produced exclusively dead embryos. (I) Typical terminal undifferentiated *fasn-1(0)* Pod embryo from *fasn-1(0); Ex[fasn-1(+)]*. ant ute, anterior uterus; post ute, posterior uterus; v, vulva; –1, most proximal oocyte. Bar, 20 μm.

The online version of this article includes the following figure supplement(s) for figure 7:

**Figure supplement 1.** Expression of *gfp::pod-2* in the germline fails to rescue *inx-8(rf)* phenotypes.

endomembrane system (*Kim et al., 2014*); reduced levels of mal-CoA in *fasn-1(0); Ex[fasn-1(+); sur-5::gfp]* could potentially reduce levels of a membrane component critical for RNA-induced silencing complex-dependent silencing of *Ex[fasn-1::gfp(+)]*. Further study is needed to address the nature of *inx-8(rf)* suppression of *fasn-1(0); Ex[fasn-1(+); sur-5::gfp]*.

## Discussion

The genetic analysis of mutant *inx-8* soma-germline gap junction channels has led us to three conclusions that have significance beyond the *C. elegans* model system. First, mal-CoA produced in the somatic sheath is delivered through gap junctions to the germline (*Figure 9*); somatically derived mal-CoA is necessary to establish a proper one-cell embryo and to support continued embryogenesis. This conclusion is supported by genetic mosaic analysis of *pod-2*, transformation rescue of *pod-2*, *fasn-1* and *emb-8* mutants, as well as genetic interactions between *inx-8(rf)* and FAS mutants. The contrasting responses of *fasn-1(g43ts)* and *emb-8(hc69ts)* to over-expression of *pod-2(+)* in the context of mutant INX-8 hemichannels, strongly support a model positioning gap junctions between *pod-2* and *fasn-1* (*Figure 9*). Furthermore, the nature of the products of these enzymes is consistent with such an interpretation. Mal-CoA produced by POD-2 is a hydrophilic, membrane impermeable molecule of a size (854 daltons) which is well within the range known to traverse the aqueous pore size of innexin gap junctions (~20–30 Å or up to ~2500 daltons; *Skerrett and Williams, 2017*). In contrast, palmitate and downstream fatty acids can cross cell membranes (*Schwenk et al., 2010*). Indeed, palmitate added to the media was shown to rescue one-cell polarity, but not osmotic integrity, in *pod-2(ye60cs)* embryos (*Rappleye et al., 2003*), suggesting that dietary palmitate can traffic from the intestine to the gonad; this result also supports a special germline role for *fasn-1* that is not complemented by dietary palmitate.

Second, *pod-2* expression (and presumably mal-CoA production) is exceptionally robust in the intestine and essential for normal germline development. In the absence of intestinal POD-2, the germline exhibits phenotypic hallmarks of the germline ARD response despite the presence of a replete food source. By contrast, intestinal *fasn-1* expression is not similarly required. (A possible caveat is that maternal *fasn-1* in the intestine may be sufficient to reach adulthood, which would be surprising.) In light of the abundant expression level of *gfp::pod-2* in the intestine and our finding that mal-CoA can be directionally transferred intercellularly, the question arises as to whether the intestine provides mal-CoA to other cells. At least one additional route may deliver mal-CoA to the germline early in its development—the primordial germ cells Z2 and Z3 associate with a pair of intestinal cells during embryogenesis, but these contacts are lost before hatching (*Sulston et al., 1983*; *Abdu et al., 2016*). Therefore, an early germline requirement for mal-CoA could be fulfilled directly from the intestine; loss of such a function would be encompassed within the ARD-like phenotype seen in *pod-2* E(–) mosaics. Potentially, mal-CoA produced in the intestine may augment levels in the somatic gonad as well as other tissues, perhaps via specific transporters or vesicles, a possibility we have not explored but which might be investigated using genetic mosaics or an auxin-inducible degron engineered into *pod-2* (*Zhang et al., 2015*). Reduced mal-CoA production in the intestine might be interpreted globally in the worm and contribute to a systemic starvation response. Recent results show that the HLH-30/TEFB transcription factor functions as a master regulator of the ARD state (*Gerisch et al., 2020*). *hlh-30* mutants fail to effect the system-wide changes promoting long-term survival and recovery associated with ARD. ARD longevity does not appear to rely on fatty acid metabolism. However, entry of *hlh-30* mutants into the ARD state appears normal. If decreased mal-

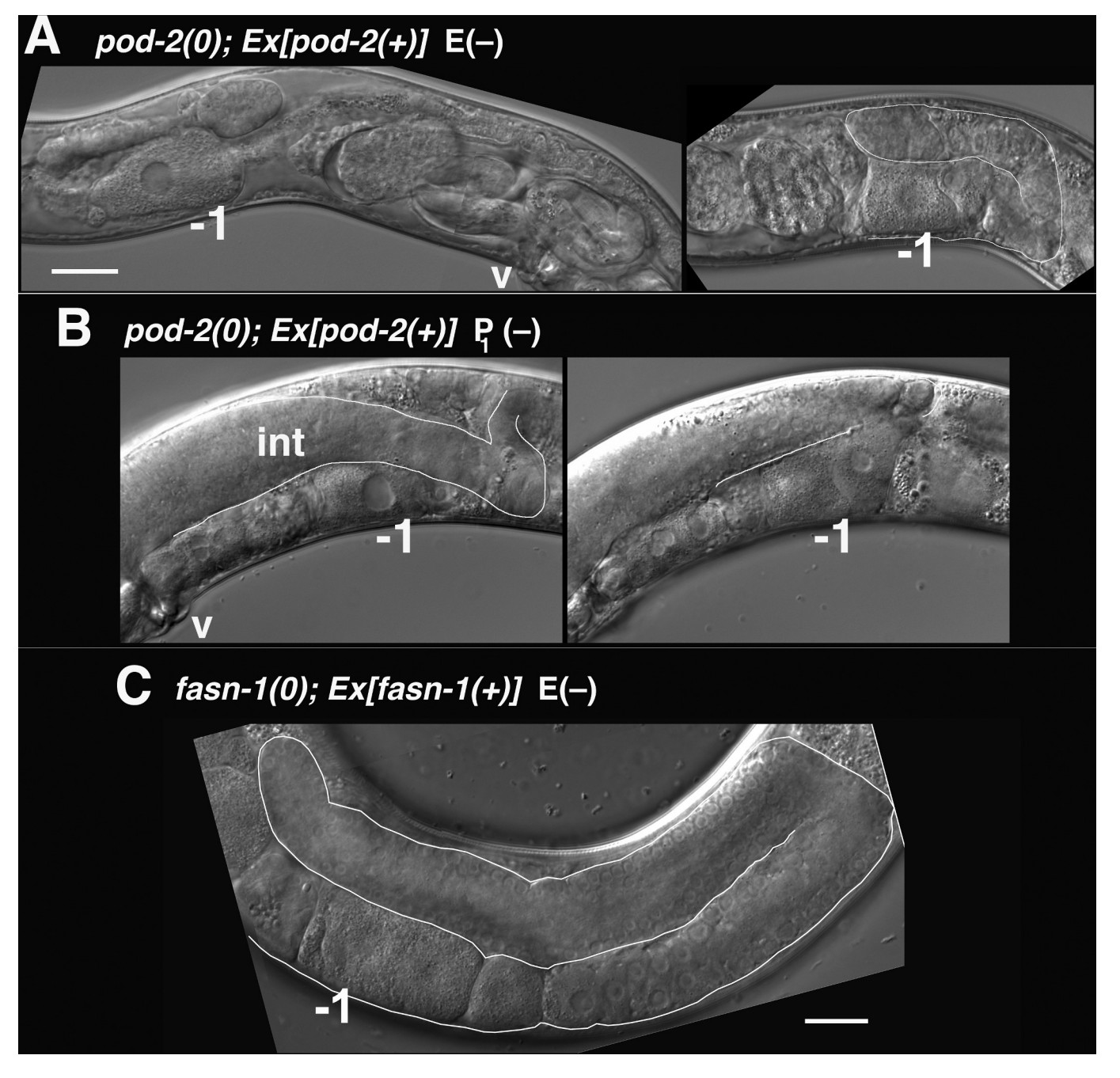

**Figure 8.** Mosaic loss from the intestine of *pod-2*, but not *fasn-1*, leads to an ARD-like gonad arm. (A) E(–) genetic mosaic *pod-2(0); Ex[pod-2(+); sur-5:: gfp]* gonad arms (from same animal). Gonad arm on right partially outlined. (B) Two focal planes of a presumptive $P_1$(–) *pod-2(0); Ex[pod-2(+); sur-5::gfp]* mosaic with constipated intestine. Upper plane (left) with partially outlined intestine; lower plane (right) shows underlying distal gonad arm. int, intestine; p, pachytene region; v, vulva. (C) E(–) genetic mosaic *fasn-1(0); Ex[fasn-1(+); sur-5::gfp]* gonad arm. –1, most proximal oocyte. Bars, 20 μm.

CoA expression in the intestine is part of the worm's initial response to starvation, such a response would lie upstream of *hlh-30* activity.

Third, a pronounced delay in gametogenesis along with reduction in brood size results from reduced soma-germline gap junction communication; it is unknown if this delay derives from generally slower mitotic or meiotic progression, or from a developmental block at a particular meiotic stage. The nature of potential molecules passing through gap junctions that are responsible for

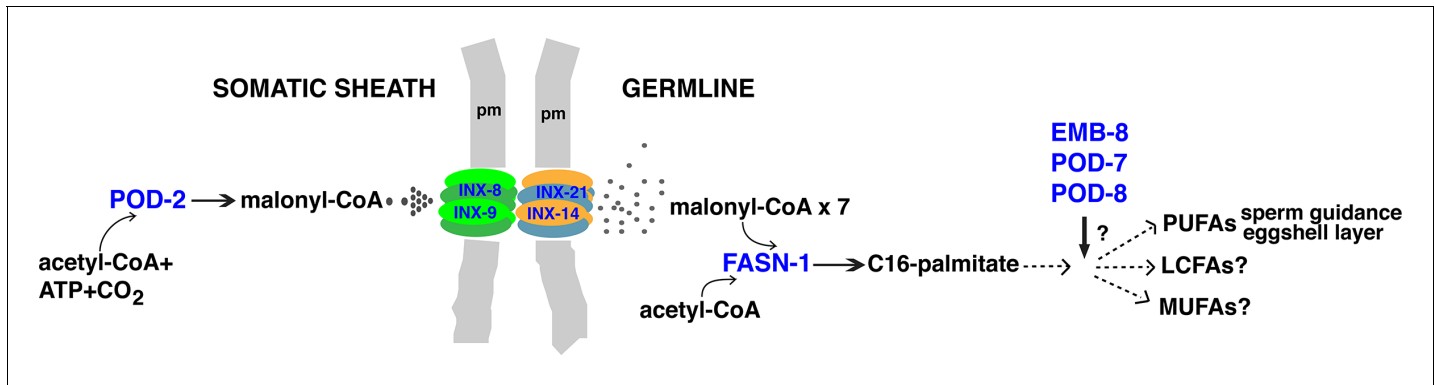

**Figure 9.** Model for malonyl-CoA transfer from soma to germline. Enzymes (blue) implicated in germline FAS. Malonyl-CoA transits from somatic sheath to germline through gap junction channels comprised of INX-8/9 (soma) and INX-14/INX-21 (germline) hemichannels. Germline FASN-1 uses malonyl-CoA to synthesize palmitate, which may be further extended and modified to produce other long-chain fatty acids. EMB-8 (NADPH-dependent CYP450 reductase) and POD-7/8 (CYP450s) are necessary to synthesize the lipid-rich (inner) layer of the eggshell and support polarity establishment (*Rappleye et al., 2003*; *Benenati et al., 2009*); the composition of lipid layer fatty acids is unknown. de novo PUFA synthesis is required in the germline for embryonic viability and is implicated in sperm guidance (*Kubagawa et al., 2006*; *Watts et al., 2018*). The identities of other long-chain fatty acids that may be synthesized in the germline are unknown. PUFAs, polyunsaturated fatty acids; LFAs, long-chain fatty acids; MUFA, monounsaturated fatty acids; pm, plasma membrane.

timely meiotic progression is unknown, but our evidence thus far has not specifically implicated mal-CoA.

Our results indicating that somatic mal-CoA transits from somatic sheath to the germline via gap junctions are consistent with other findings in *C. elegans* related to fatty acid metabolism in oocytes, particularly a strict requirement for PUFA synthesis (*Watts et al., 2018*). PUFA-dependent prostaglandin signaling from developing oocytes influences sperm guidance, and reduction of *inx-14* function disrupts this signaling (*Kubagawa et al., 2006*; *Edmonds et al., 2011*). We suggest that lower levels of mal-CoA delivered to the oogenic germline through mutant INX-14-containing gap junctions reduce prostaglandin synthesis and its signaling function for sperm migration.

To our knowledge, this segregation of the cellular source of mal-CoA from the eventual cell type in which it is used as substrate is novel. As the rate-limiting step in FAS, mal-CoA production is an attractive control point to tie aspects of germ cell developmental progression to somatic control. However, although genetic evidence indicates a germline role for *fasn-1*, the inability to visualize GFP-tagged FASN-1 suggests maternal *fasn-1* is expressed at very low levels. We therefore consider whether mal-CoA could function in the germline outside the standard FAS pathway, a possibility we attempted to examine using embryo output as a parameter.

Cytoplasmic FAS is carried out by a multifunctional megasynthase like that encoded by *fasn-1* (type-I system). Mitochondria carry out FAS independently using monofunctional enzymes resembling the bacterial FAS type-II system (recently reviewed by *Kastaniotis et al., 2017*). Mitochondrial FAS utilizes an acyl-carrier protein rather than CoA to shuttle the substrate between enzymes. An ACC enzyme has been found in the mitochondria of *S. cerevisiae* but not mammals. In mammals, the ACSF3 enzyme uses malonate as a substrate to synthesize mal-CoA and could contribute to mitochondrial FAS. However, malonate is a toxic metabolite that inhibits succinate dehydrogenase, and knockout of ACSF3 in cultured cells suggests this enzyme is more critical for eliminating toxic malonate from mitochondria rather than serving an essential function for mitochondrial FAS (*Bowman et al., 2017*). The sources of malonate and mal-CoA in mitochondria are still unclear, although it has been suggested that hydrolysis of cytosolic mal-CoA may produce malonate that is taken up by mitochondria (*Bowman et al., 2017*). If this is the case, reduced levels of cytoplasmic mal-CoA could negatively impact mitochondrial FAS.

In addition to its metabolic function in FAS, mal-CoA has established roles as a regulatory molecule. It inhibits uptake and therefore oxidation of fatty acyl-CoAs by carnitine palmitoyltransferase I (CPT-1). Beta-oxidation of fatty acids in mitochondria provides energy through oxidative phosphorylation. The inhibitory mal-CoA in some tissues that exhibit low levels of FAS (heart and skeletal) derives from a second acetyl-CoA carboxylase (ACC2), which associates with the mitochondrial

membrane to produce mal-CoA regulating CPT-1. Manipulation of mal-CoA levels in the mouse hypothalamus regulates food intake (*Wolfgang and Lane, 2006*). Regulation is effected through the brain-specific CPT1-c, via a mechanism distinct from the enzymatic activity of other CPT1 enzymes (*Sierra et al., 2008*; *Reilly and Mak, 2012*). Little information regarding the *C. elegans* ACC2 is available. Therefore, *pod-2*/ACC type I-derived mal-CoA delivered from the somatic sheath could in effect be relaying the state of nutrition to the germline and modulating regulation of fatty acid beta-oxidation in mitochondria accordingly.

Malonylation as a post-translational lysine modification has recently been described (*Peng et al., 2011*). Malonylation may inhibit target proteins, such as the glycolytic enzymes glyceraldehyde 3-phosphate dehydrogenase and pyruvate kinase (*Kulkarni et al., 2017*), and ACSF3 was shown to be essential for malonylation of mitochondrial protein targets (*Bowman et al., 2017*). Malonylation of mTOR reduces mTORC1 complex activity, decreasing endothelial cell proliferation and reducing angiogenesis in a human umbilical model (*Bruning et al., 2018*). Together these studies and others suggest a possibly pervasive role for mal-CoA in sensing and responding to nutritional status. Levels of mal-CoA provided to the *C. elegans* germline from the soma could potentially regulate a number of metabolic processes in both mitochondria and the cytoplasm.

Finally, we note that one trait associated with rapid tumor growth is an increase in FAS, and FASN has been explored as a cancer therapy target (*Menendez and Lupu, 2007*). Possibly, the ability of mal-CoA to transit through gap junctions might sustain FAS during tumor progression. The discovery that mal-CoA transits through gap junctions from soma to germline may be indicative of other unappreciated intercellular roles for this important molecule.

## Materials and methods

### Strains

*C. elegans* strains were grown on standard NGM media (plus Nystatin, 6.25 mg/l) with *E. coli* strain OP50 as food source. Temperatures used for conditional mutants are indicated in main text. In addition to the wild-type strain N2, the following alleles were used: Chr. *I: fasn-1(g43ts); fasn-1 (tn1762null); fasn-1(tn1782[gfp::fasn-1]); fasn-1(av138[fasn-1::gfp]); lin-41(tn1541[gfp::tev::s-tag::lin-41]);* Chr. *II: pod-2(ye60cs); pod-2(tn1691null); pod-2(tn1765[gfp::tev::3xFLAG::pod-2]);* Chr. *III: emb-8(hc69ts); dpy-17(e1295);* Chr. *IV: inx-8(tn1474null) inx-9(ok1502null); inx-8(tn1513) inx-9(ok1502); inx-8(tn1513 tn1555) inx-9(ok1502)*—a.k.a. *inx-8(rf)* in main text. Balancers used included: *tmC18 [dpy-5(tmIs1236)] I; hT2[bli-4(e937) let-?(q782) qIs48] (I;III); mIs11 IV*. Extrachromosomal arrays used included: *tnEx195[inx-8(+) inx-9(+); sur-5::gfp]; tnEx205[lag-2p::inx-8::gfp; str-1::gfp]; tnEx212 [W09B6-pod-2(+)—20 ng/μl—hi copy; sur-5::gfp]; tnEx218[fasn-1(+); sur-5::gfp]; tnEx219[W09B6-pod-2(+)—4 ng/μl—lo copy; sur-5::gfp];* and *tnEx221[inx-8(tn1513 tn1555); str-1::gfp].* *Supplementary file 1* contains a full list of all strains used in this study.

### Microinjection rescue of FAS mutants with extrachromosomal arrays

The cosmid W09B6 was used at 50 ng/μl (two lines) or 20 ng/μl (one line) with *sur-5::gfp* (75 ng/μl) to rescue *pod-2(ye60cs)*. *sur-5::gfp* expression was faint for both lines obtained with 50 ng/μl and these were not further characterized. *pod-2(tn1691null)* was rescued by microinjection using W09B6 at 4 ng/μl (five lines) or by crossing in *tnEx212*. Fosmids WRM0613bD07 (10 ng/μl, or 50 ng/μl—one line that silenced) and WRM067cB03 (25 ng/μl) were used to attempt rescue of *emb-8(hc69ts)*. Numerous conditions using fosmids WRM0612bD08 and WRM0614aH05 in simple or complex arrays were used to try to rescue *fasn-1(g43ts)* without success. However, FASN enzyme functions as a dimer; formation of dimers between mutant (*g43*) and wild-type monomers might inhibit phenotypic rescue. After *fasn-1(tn1762null)* was generated, somatic but not germline rescue was obtained using a *fasn-1* genomic PCR product forming simple arrays (1 ng/μl + *sur-5::gfp*-75 ng/μl, at least 2/6 lines segregated *fasn-1(0); Ex[fasn-1(+); sur-5::gfp(+)]* animals that laid only Pod embryos), or a complex array (1 ng/μl + 20 ng/μl *sur-5::gfp* + 100 ng/μl salmon sperm,1 line from 30 injected animals) as described in the main text.

## Recombinant DNA constructs for isolation of new *pod-2* and *fasn-1* alleles

Standard CRISPR/Cas-9 methods were carried out in the wild-type Bristol N2 strain to generate deletions and GFP insertions in *pod-2* and *fasn-1*. For deletions, 100-nt repair templates were synthesized (Ultramers; Integrated DNA Technologies, Coralville, IA) and used in a *dpy-10(cn64)* co-conversion strategy previously described (*Arribere et al., 2014*). Injection mixes included: pDD162 (Cas9), 50 ng/μl; 25 ng/μl (*pod-2*) or 10 ng/μl (*fasn-1*) each single guide; Ultramer repair template, 500 nM; *dpy-10* single guide (pJA58), 25 ng/μl; *dpy-10(cn64)* Ultramer A2-2F-827, 500 nM. Single guides were made by annealing oligo pairs and cloning into pRB1017 (*Arribere et al., 2014*). Deletions were verified by PCR. The PCR fragment used to rescue *fasn-1(tn1762null)* was amplified from genomic DNA using primers 5'-tacgtcaaacatccgtttgtcaacgtcac-3' and 5'-ttcttctcctcgtcctttgatacgaagatc-3'.

Generation of GFP insertions into *pod-2* using pDD282 followed established protocols (*Dickinson et al., 2015*). Injection mix included the single guide (10 ng/μl), repair template (10 ng/μl), pDD162 (50 ng/μl), and *sur-5::gfp* (80 ng/μl). The insertion was verified by sequencing PCR products generated with primers positioned outside the homology arms.

For *gfp::fasn-1* we used a co-conversion approach (*Arribere et al., 2014*). A repair template was generated that included the GFP sequence from pDD282, and *fasn-1* homology arms on either side. The injection mix contained pDD162, 50 ng/μl; *fasn-1* single guide 10 ng/μl; *gfp::fasn-1* repair template, 20 ng/μl; *dpy-10* single guide (pJA58), 25 ng/μl; *dpy-10(cn64)* Ultramer A2-2F-827, 500 nM; and pCFJ190, 2.5 ng/μl. Rol candidates were screened directly for GFP expression. The insertion was verified by sequencing PCR fragments generated using primers placed outside of the repair template sequence.

To generate *fasn-1::gfp*, the *fasn-1* specific 20-nucleotide sequence for crRNA was selected with help of a guide RNA design checker from Integrated DNA Technologies (IDT) (https://www.idtdna.com) and was synthesized as a 20 nmol product from Dharmacon (https://dharmacon.horizondiscovery.com), along with tracrRNA. The repair template was amplified from plasmid pDD282 (https://www.addgene.org/66823/) and designed to include a flexible linker (GASGASGAS) and GFP at the C-terminus of the *fasn-1* endogenous locus. These were designed using a standard protocol (*Paix et al., 2015*). Approximately 30 young gravid animals were injected with the CRISPR/Cas9 injection mix as described in the standard protocol. Homozygous genome-edited animals were screened by PCR and confirmed by Sanger sequencing (Eurofins).

*mex-5p::gfp::pod-2::tbb-2–3'UTR* was constructed by Gibson assembly of component PCR products representing the *mex-5* promoter, the 'a' isoform of *gfp::pod-2*, and the *tbb-2–3'UTR* to prevent germline RNA degradation.

## Immunofluorescence

Antibody staining of dissected gonads was as previously described (*Starich et al., 2014*). DIC and fluorescent images were acquired on a Zeiss (Thornwood, NY) motorized Axioplan 2 microscope with either a 40x Plan-Neofluar (numerical aperture 1.3), a 63x Plan-Apochromatic (numerical aperture 1.4), or 100x PlanApochromatic (N.A. 1.4) objective lens using an AxioCam MRm camera and AxioVision software (Zeiss).

For *fasn-1(av138[fasn-1::gfp])* imaging data, animals were immobilized on 7% agarose pads with 0.05 μm polystyrene beads and imaged using a spinning disk confocal system with a Nikon 60 × 1.2 NA water objective, a Photometrics Prime 95B EMCCD camera, and a Yokogawa CSU-X1 confocal scanner unit. Images were acquired and analyzed by Nikon's NIS imaging software and ImageJ/FIJI Bio-formats plugin (National Institutes of Health) (*Linkert et al., 2010*; *Schindelin et al., 2012*). The acquisition of GFP and DIC images was performed with 1 μm z-step size and 15–20 z planes.

## Genetic mosaic analysis

To associate the somatic focus of action of *pod-2* with germline phenotypes, array losses were characterized in animals of genotype *pod-2(tn1691null); tnEx212[W09B6-pod-2(+)—20 ng/μl—hi copy; sur-5::gfp]* that were isolated on the basis of sterility. Subsequently, potential candidate mosaics of interest were prescreened directly at the dissecting microscope level for asymmetry in the phenotypic severity seen between anterior and posterior gonad arms, and then examined at the

compound microscope level for loss of *sur-5::gfp* expression. Potential *pod-2(tn1691null); tnEx219 [W09B6-pod-2(+)—4 ng/μl—lo copy; sur-5::gfp]* genetic mosaics were also prescreened in this manner. Intestinal (E–) mosaics were prescreened using a low-power fluorescence SMZ18 Nikon (Melville, NY) microscope with a Lumencor (Beaverton, OR) SOLA Light Engine, but for *pod-2* many of these candidate mosaics were seen to express *sur-5::gfp* at very low levels, or in just a few intestinal cells, when examined at high power. This was not an issue when isolating *fasn-1(tn1762null); tnEx218[fasn-1(+); sur-5::gfp]* E(–) mosaics.

## Oligos used

### *fasn-1(tn1762)* isolation
Single guide pairs:

> 5'-tcttggtatcttcccactcactga-3' plus 5'-aaactcagtgagtgggaagatacc-3', and
> 5'-cttggatgactttccttgcacga-3' plus 5'-aaactcgtgcaaggaaagtcatcc-3'.

Repair template:

> 5'cgtatgacgtgattttaatgtgataattactctgaaatatctcaaaacgtgcaaggaaagtcatcttcagtaacagttgaaca-catcaacaggattat-3'

Verification of *tn1762*:

> 5'-ctgacaagacgacggacaactcagg-3' and 5'- ctacgtaaccgtaacggcatggcac-3'

### *pod-2(tn1691)* isolation
Single guide pairs:

> 5'-tcttggaaaactgtccatatacaa-3' plus 5'-aaacttgtatatggacagttttcc-3', and
> 5'-tcttgacagaacaggaaaaagtgg-3' plus 5'-aaacccagttttttcctgttctgtc-3'.

Repair template:

> 5'-ccgattttttgtgcaatttcagagcaatataagtatataaacgttattttcagagccatcacttttcctgttctgtgatatacaa-gaaacctattcagtgccttatttgattgacgactac-3'

Verification of *tn1691* employed oligos:

> 5'-cgacgaagtaagtgagcctaattgtac-3' with 5'-ttgtcaaccaccttacagtggcatg-3'

### *gfp::pod-2(tn1765)* isolation
Single guide pair:

> 5'-tcttgcagggttacggatcttgga-3' and 5'-aaactccaagatccgtaaccctgc-3

Homology arm oligo pairs:

> 5'-acgttgtaaaacgacggccagtcgccggcatctacctgtgtacctgacctgaccagattc-3' plus
> 5'-aactccagtgaacaattcttctcctttactcattgtgaatgccagtccaagatccgtaac-3';
> 5'-cgtgattacaaggatgacgatgacaagagagttgtaaacgggcaaaaaccagatatcag-3' plus
> 5'-tcacacaggaaacagctatgatatgtgagagtgaatgaactgctccatgctctc-3'

### *gfp::fasn-1(tn1782)* isolation
Single guide pair:

> 5'-tcttgaaggatcgtccgattcgag-3' plus 5'-aaaactcgaatcggacgatccttc-3'

Primer used to introduce silent mutations to recognition site:

> 5'-gataatactggagagggctcatcagactcaagtggaacttgggagcgaatttcggac-3'

Homology flanking GFP:

> 5'-gggataataagtaggttctgacctcctctcccg-3' to 5'-gatccattcgtgcttcttcttgtcccgtgag-3'

*fasn-1::gfp(av138)* isolation

crRNA C-terminus:

>5' TAACAGTTGAACACATCAAC 3'

PAM site:

>AGG

Primer Repair Template F1:

>(5'-3') AAGGAAAGTCATCTTCAGTAACAGTTGAACACATCAACAGAATCATACTGCAG *GGAGCATCGGGAGCC (pDD282 plasmid)*

Primer Repair Template R1:

>(5'- 3') TTGGATGAGATTATGGATATGTGATTGTTGATTTA *CTTGTAGAGCTCGTCCATTC (pDD282 plasmid)*

Primer Genotype F1:

>(5'- 3') ACTTCGGTGGACACATCACA

Primer Genotype R1:

>(5'- 3') TGAGTCTTGAGCAAAGAGCA

## Acknowledgements

We are grateful to our colleagues Gabriela Huelgas-Morales, Caroline Spike, Tatsuya Tsukamoto, and Jocelyn Shaw for discussions regarding experiments and comments on the manuscript. We thank Andy Golden for his guidance and generosity during the course of this work. We thank Ross Johnson and David Hall for many years of sharing their knowledge and enthusiasm in all things concerning gap junctions. This work was supported by National Institutes of Health (NIH) grant GM57173 to DG. This work was also supported in part by the Intramural Research Program of the National Institutes of Health, National Institute of Diabetes and Digestive and Kidney Diseases (XB, who is a member of Dr. Andy Golden's laboratory). Some strains were provided by the Caenorhabditis Genetics Center, which is funded by grant P40OD010440 from the NIH Office of Research Infrastructure Programs.

## Additional information

### Funding

| Funder | Grant reference number | Author |
| --- | --- | --- |
| National Institutes of Health | GM57173 | David Greenstein |

The funders had no role in study design, data collection and interpretation, or the decision to submit the work for publication.

### Author contributions

Todd A Starich, Conceptualization, Resources, Data curation, Formal analysis, Supervision, Funding acquisition, Validation, Investigation, Visualization, Methodology, Writing - original draft, Project administration, Writing - review and editing; Xiaofei Bai, Formal analysis, Validation, Investigation, Visualization, Methodology, Writing - review and editing; David Greenstein, Conceptualization, Supervision, Funding acquisition, Validation, Investigation, Methodology, Project administration, Writing - review and editing

### Author ORCIDs

Xiaofei Bai http://orcid.org/0000-0001-8179-8162

David Greenstein (iD) https://orcid.org/0000-0001-8189-2087

**Decision letter and Author response**
Decision letter https://doi.org/10.7554/eLife.58619.sa1
Author response https://doi.org/10.7554/eLife.58619.sa2

## Additional files

### Supplementary files
- Supplementary file 1. *C. elegans* strains used in this study.
- Transparent reporting form

### Data availability

All *C. elegans* strains are available from the *Caenorhabditis* Genetics Center or by request. All data supporting the findings of this study are contained within the manuscript, figures tables, or source data provided.

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
