## [Decision Letter]

**Acceptance summary:**

Using *C. elegans* as a model system, the authors provide strong genetic evidence that malonyl-CoA, produced in somatic cells, is delivered to the germline through gap junctions. Phenotypic characterization of several mutants within this pathway suggest that the transport of metabolites through gap junctions may play a regulatory role by which the soma communicates nutritional status to the germline.

**Decision letter after peer review:**

Thank you for submitting your article "Gap junctions deliver malonyl-CoA from soma to germline to support embryogenesis in *Caenorhabditis elegans*" for consideration by *eLife*. Your article has been reviewed by three peer reviewers, including Michael Buszczak as the Reviewing Editor and Reviewer #1, and the evaluation has been overseen by Piali Sengupta as the Senior Editor.

The reviewers have discussed the reviews with one another and the Reviewing Editor has drafted this decision to help you prepare a revised submission.

In this manuscript, Starich and Greenstein present data that malonyl-CoA, produced in somatic cells, is delivered to the germline through gap junctions. Further understanding communication between somatic cells and the germline, and uncovering new mechanisms that regulate germline development in response to systemic cues is of significant interest to the field of reproductive biology. This study provides solid genetic evidence, including the characterization of new alleles and mosaic analysis, that gap junctions represent a functionally important line of communication between somatic cells and the germline.

Summary:

The study starts with the isolation of an intragenic suppressor of a strong *inx-8* hypomorphic allele. The resulting allele (*inx-8(rf)*) displays fertility at one third the level of wild-type and an 18 hour delay in fertilization. Next, antibodies against INX-8 and two germline innexin proteins (INX-21 and INX-22) were used to stain wild-type samples and different mutants. This analysis revealed that *inx-8(rf)* displayed little gap junction coupling in proximal arms of the gonad. Further genetic analysis revealed interactions between *inx-8(rf)* and Pod (Polarity and Osmotic sensitivity Defect) mutants, including *fasn-1* (fatty acid synthase) and *emb-8* (NADPH-cytochrome-P450 reductase). Mosaic analysis suggested that *pod-2* acts in the somatic gonad, whereas *fasn -1* functions in the germline. Endogenously tagged *pod-2* and *fasn-1* showed that both genes exhibited a tissue specific expression pattern, but neither could be visually detected in the germline. However, genetic tests indicated that low levels of germline FASN-1 must be present. The authors then reason that if INX-8 is needed for the transport of Mal-CoA (product of POD-2) from the soma to the germline, then the fertility of mutants downstream of *pod-2* could be compromised by INX-8(rf). This is indeed what they find. Further mosaic analysis indicates that *pod-2* loss in the sheath cell can account for many aspects of the Pod phenotype. In addition, loss of *pod-2* in the intestine, but not *fasn-1*, results in a germline starvation phenotype. Attempts to rescue inx8/9 double mutant phenotypes by microinjection of mal-CoA into gonads failed. All together these results lead the authors to make three conclusions: (1) Mal-CoA produced in the somatic sheath is delivered through gap junctions to the germline, (2) *pod-2* expression is high in the intestine and necessary for normal germline development, and (3) reduced soma-germline communication causes a delay in gametogenesis and reduced brood size.

Essential revisions:

1) In general, the writing could be more accessible. It may be especially difficult for a non-worm audience. The story is somewhat complex with several arguments and it may be difficult for the reader to connect the dots without additional guidance.

2) The presentation and the findings used to indicate that it is malonyl-CoA that is transported through the gap junctions and that *fasn-1* functions in the germline lead to a weaker case. The *fasn-1* knock-in GFP expression studies shows no evidence of germline expression, but clear somatic gonadal sheath expression. While it is known that germline expressed extra-chromosomal transgenes are often silenced and thus do not rescue, this is a negative result. Positive data are that *fasn-1* and *emb-8* phenotypes, which are modified by *Ex[pod-2(+)]* (that likely results in excess malonyl-CoA), show genetic dependency with *inx-8* activity, and that *fasn-1* activity is maternally provided for early embryonic Pod functions.

While orthogonal data for *fasn-1* accumulation/function in the oocyte would certainly be helpful, changes in presentation would help make a stronger case. In this Discussion, it would be helpful to first summarize the findings that *pod-2* functions/malonyl-CoA is synthesized in sheath cells and second to summarize the data that suggests that *fasn-1* functions in the germline and thus malonyl-CoA is transported through gap junctions. Then it would be useful to discuss the size and type of molecules (hydrophilic?) that are known to be transported through gap junctions. It would seem that if *fasn-1* functioned in sheath cells, it would be problematic for molecules like palmitate to be transferred to oocytes through gap junctions. This would then provide the context for the presentation of many of the ideas in the current Discussion. Additional recommended changes to the text are indicated below.

3) The intestinal loss of *pod-2* results in an adult reproductive diapause like phenotype. However, the authors need to be careful not to over-sell the finding as there are no experiments/results that specifically tie the observed phenotype to starvation/replete environment and genes known to function in ARD.

4) Figure 6C should be discussed/presented as a model.

5) The observations that starvation decreases POD-2::GFP expression and loss of *pod-2* in the intestine phenocopies ARD are particularly intriguing and possibly related. But neither observation is elaborated upon in the text.

---

## [Author Response]

Essential revisions:1) In general, the writing could be more accessible. It may be especially difficult for a non-worm audience. The story is somewhat complex with several arguments and it may be difficult for the reader to connect the dots without additional guidance.

We have tried to be more explicit and flesh out some of our conclusions in building the argument that malonyl-CoA transits from soma to germline through gap junctions. We have extensively rewritten the Introduction and the Results section to make them more reader friendly for a general audience. In the absence of being able to physically see mal-CoA transit between cells, we must rely on genetic evidence. Our initial evidence regarding rescue is supportive, but certainly not conclusive. However, we feel that the experiments presented in Figure 6 are at the crux of the matter – we cannot offer any other models that would reconcile these data. We admit that these experiments require some thought – we ourselves have to work through these relationships mentally step by step even when going back to look at the data. We have added what we hope are visual aids in Figure 6 (new parts C and D) that readers can look at when considering the data – hopefully this more concrete presentation of our interpretation of what is happening in parts A and B will help readers follow our thought process.

2) The presentation and the findings used to indicate that it is malonyl-CoA that is transported through the gap junctions and that fasn-1 functions in the germline lead to a weaker case. The fasn-1 knock-in GFP expression studies shows no evidence of germline expression, but clear somatic gonadal sheath expression. While it is known that germline expressed extra-chromosomal transgenes are often silenced and thus do not rescue, this is a negative result. Positive data are that fasn-1 and emb-8 phenotypes, which are modified by Ex[pod-2(+)] (that likely results in excess malonyl-CoA), show genetic dependency with inx-8 activity, and that fasn-1 activity is maternally provided for early embryonic Pod functions.While orthogonal data for fasn-1 accumulation/function in the oocyte would certainly be helpful, changes in presentation would help make a stronger case. In this Discussion, it would be helpful to first summarize the findings that pod-2 functions/malonyl-CoA is synthesized in sheath cells and second to summarize the data that suggests that fasn-1 functions in the germline and thus malonyl-CoA is transported through gap junctions. Then it would be useful to discuss the size and type of molecules (hydrophilic?) that are known to be transported through gap junctions. It would seem that if fasn-1 functioned in sheath cells, it would be problematic for molecules like palmitate to be transferred to oocytes through gap junctions. This would then provide the context for the presentation of many of the ideas in the current Discussion. Additional recommended changes to the text are indicated below.

We thank the reviewers for this insightful suggestion, and we have reorganized both our Introduction and Discussion. (Although we would like to point out that the evidence that *fasn-1* is expressed in the germline does not derive solely from negative data, i.e. the assumption that failure to rescue indicates germline silencing. This interpretation is also supported by *inx-8(rf)* partially rescuing *fasn-1; Ex[fasn-1(+); sur-5::gfp]* sterility, suggesting that *fasn-1* on this array can be expressed in the germline under certain conditions. The alternative explanation would be that somehow compromising soma–germline gap junction coupling in *inx-8(rf)* allows somatically-expressed FASN-1 to rescue Pod embryos.)

We have moved much of our description of FAS gene mutants that produce Pod embryos to the Introduction to better prepare the reader for why we examined genetic interactions between *inx-8* and FAS mutations. We have then included a brief description in the Discussion of the molecular nature of mal-CoA and palmitate and their respective “suitability” for transmission through gap junctions, as suggested.

3) The intestinal loss of pod-2 results in an adult reproductive diapause like phenotype. However, the authors need to be careful not to over-sell the finding as there are no experiments/results that specifically tie the observed phenotype to starvation/replete environment and genes known to function in ARD.

We thank the reviewer for the suggestion. In the reworked text, we have been careful not to over-sell these findings. Although there are currently no molecular markers specific for the ARD response, our images of the *pod-2* E(–) genetic mosaics are essentially identical to what was previously reported for the ARD response under starvation conditions (Angelo and Van Gilst, 2009; Seidel and Kimble, 2011). Thus, the function of *pod-2* in the intestine appears to be required to inhibit the ARD response under replete conditions. This is analogous to genes in the insulin pathway, which inhibit dauer formation under replete conditions. We feel the field will be very interested in this finding and we think our treatment in the revised manuscript is appropriately balanced. See response to point #5, below for additional related comments.

4) Figure 6C should be discussed/presented as a model.

This was a good suggestion. We have moved what was Figure 6C to its own Figure 9 and referred to it in the Discussion.

5) The observations that starvation decreases POD-2::GFP expression and loss of pod-2 in the intestine phenocopies ARD are particularly intriguing and possibly related. But neither observation is elaborated upon in the text.

As per point #3 above, we tried to avoid speculating too much about these results but welcome the reviewer’s invitation to discuss them. The fact that *pod-2* is expressed at very high levels in the intestine and is responsive to starvation, while *fasn-1* is undetectable, was quite surprising. We have now included *fasn-1::gfp* expression data from our added co-author, Xiaofei Bai, which has given us more confidence that *fasn-1* is not expressed in the intestine at detectable levels (we can rule out additional isoforms because we have now tagged *fasn-1* at both ends). As far as we know this is an unprecedented situation–two enzymes in a linear metabolic pathway are expressed and function in different cells! Therefore, our speculations cannot borrow from any similar examples. We have added to the Discussion possible implications of intestinal production of mal-CoA not destined for use by the FAS pathway in the intestine. Regarding point #3 above, we have speculated on how starvation may lead to a response in the intestine through *pod-2* that may be part of a greater starvation response, without claiming a direct connection to the currently proposed ARD pathway.